# THE EXPRESSIVE POWER OF GATED RECURRENT UNITS AS A CONTINUOUS DYNAMICAL SYSTEM

## ABSTRACT

Gated recurrent units (GRUs) were inspired by the common gated recurrent unit, long short-term memory (LSTM), as a means of capturing temporal structure with less complex memory unit architecture. Despite their incredible success in tasks such as natural and artificial language processing, speech, video, and polyphonic music, very little is understood about the specific dynamic features representable in a GRU network. As a result, it is difficult to know a priori how successful a GRU-RNN will perform on a given data set. In this paper, we develop a new theoretical framework to analyze one and two dimensional GRUs as a continuous dynamical system, and classify the dynamical features obtainable with such system. We found rich repertoire that includes stable limit cycles over time (nonlinear oscillations), multi-stable state transitions with various topologies, and homoclinic orbits. In addition, we show that any finite dimensional GRU cannot precisely replicate the dynamics of a ring attractor, or more generally, any continuous attractor, and is limited to finitely many isolated fixed points in theory. These findings were then experimentally verified in two dimensions by means of time series prediction.

## 1 INTRODUCTION

Recurrent neural networks (RNNs) have been widely used to capture and utilize sequential structure in natural and artificial languages, speech, video, and various other forms of time series. The recurrent information flow within RNN implies that the data seen in the past has influence on the current state of the RNN, forming a mechanism for having memory through (nonlinear) temporal traces. Unfortunately, training vanilla RNNs (which allow input data to directly interact with the hidden state) to capture long-range dependences within a sequence is challenging due to the vanishing gradient problem (Hochreiter, 1991). Several special RNN architectures have been proposed to mitigate this issue, notably the long short-term memory (LSTM; Hochreiter & Schmidhuber (1997)) which explicitly guards against unwanted corruption of the information stored in the hidden state until necessary. Recently, a simplification of the LSTM called the *gated recurrent unit* (GRU; Cho et al. (2014)) has become wildly popular in the machine learning community thanks to its performance in machine translation (Britz et al., 2017), speech (Prabhavalkar et al., 2017), music (Choi et al., 2017), video (Dwibedi et al., 2018), and extracting nonlinear dynamics underlying neural data (Pandarinath et al., 2018). As a variant of the vanilla LSTM, GRUs incorporate the use of forget gates, but lack an output gate (Gers et al., 2000). While this feature reduces the number of required parameters, LSTM has been shown to outperform GRU on neural machine translation (Britz et al., 2017). In addition, certain mechanistic tasks, specifically unbounded counting, come easy to LSTM networks but not to GRU networks (Weiss et al., 2018). Despite these empirical findings, we lack systematic understanding of the internal time evolution of GRU's memory structure and its capability to represent nonlinear temporal dynamics.

In general, a RNN can be written as $\mathbf{h}_{t+1} = f(\mathbf{h}_t, \mathbf{x}_t)$ where $\mathbf{x}_t$ is the current input in a sequence indexed by $t$, $f$ is a point-wise nonlinear function, and $\mathbf{h}_t$ represents the hidden memory state that carries all information responsible for future output. In the absence of input, the hidden state $\mathbf{h}_t$ can evolve over time on its own:

$$\mathbf{h}_{t+1} = f(\mathbf{h}_t) \tag{1}$$

where $f(\cdot) := f(\cdot, \mathbf{0})$ for notational simplicity. In other words, we can consider the temporal evolution of memory stored within RNN as a trajectory of a dynamical system defined by (1). Then we can use dynamical systems theory to investigate the fundamental limits in the expressive power of RNNs in terms of their temporal features. We develop a novel theoretical framework to study the dynamical features fundamentally attainable, in particular, given the particular form of GRU. We then validate the theory by training GRUs to predict time series with prescribed dynamics.

## 2    CONTINUOUS-TIME GATED RECURRENT UNIT

The GRU uses two internal gating variables: the *update gate* $\mathbf{z}_t$ which protects the $d$-dimensional hidden state $\mathbf{h}_t \in \mathbb{R}^d$ and the *reset gate* $\mathbf{r}_t$ which allows overwriting of the hidden state and controls the interaction with the input $\mathbf{x}_t \in \mathbb{R}^p$.

$$\mathbf{z}_t = \qquad \sigma(\mathbf{W}_z\mathbf{x}_t + \mathbf{U}_z\mathbf{h}_{t-1} + \mathbf{b}_z) \qquad \textit{(update gate)} \qquad (2)$$

$$\mathbf{r}_t = \qquad \sigma(\mathbf{W}_r\mathbf{x}_t + \mathbf{U}_r\mathbf{h}_{t-1} + \mathbf{b}_r) \qquad \textit{(reset gate)} \qquad (3)$$

$$\mathbf{h}_t = (1 - \mathbf{z}_t)\odot\tanh(\mathbf{W}_h\mathbf{x}_t + \mathbf{U}_h(\mathbf{r}_t \odot \mathbf{h}_{t-1} + \mathbf{b}_h)) + \mathbf{z}_t \odot \mathbf{h}_{t-1} \quad \textit{(hidden state)} \qquad (4)$$

where $\mathbf{W}_z, \mathbf{W}_r, \mathbf{W}_h \in \mathbb{R}^{d\times p}$ and $\mathbf{U}_z, \mathbf{U}_r, \mathbf{U}_h \in \mathbb{R}^{d\times d}$ are the parameter matrices, $\mathbf{b}_z, \mathbf{b}_r, \mathbf{b}_h \in \mathbb{R}^d$ are bias vectors, $\odot$ represents the Hadamard product, and $\sigma(\mathbf{z}) = 1/(1 + e^{-\mathbf{z}})$ is the element-wise logistic sigmoid function. Note that the hidden state is asymptotically contained within $[-1, 1]^d$ due to the saturating nonlinearities, implying if the state is initialized outside this trapping region, it must eventually enter it in finite time and remain in it for all later time.

Note that the update gate $\mathbf{z}_t$ controls how fast each dimension at the hidden state decays, providing an adaptive time constant for memory. Specifically, as $\lim_{\mathbf{z}_t \to 0} \mathbf{h}_t = \mathbf{h}_{t-1}$, GRUs can implement perfect memory of the past and ignore $\mathbf{x}_t$. Hence, a $d$-dimensional GRU is capable of keeping a near constant memory through the update gate—near constant since $0 < [\mathbf{z}_t]_j < 1$, where $[\cdot]_j$ denotes $j$-th component of a vector. Moreover, the autoregressive weights (mainly $\mathbf{U}_h$ and $\mathbf{U}_r$) can support time evolving memory (Laurent & von Brecht (2016)) considered this a hindrance and proposed removing all complex dynamic behavior in a simplified GRU).

To investigate the memory structure further, let us consider the dynamics of hidden state in the absence of input, i.e. $\mathbf{x}_t = 0, \forall t$, which is of the form (1). To utilize the rich descriptive language of continuous time dynamical system theory, we consider the following continuous time limit of the (autonomous) GRU time evolution:

$$\mathbf{z}(t) = \sigma(\mathbf{U}_z\mathbf{h}(t) + \mathbf{b}_z) \qquad \textit{(continuous update gate)} \qquad (5)$$

$$\mathbf{r}(t) = \sigma(\mathbf{U}_r\mathbf{h}(t) + \mathbf{b}_r) \qquad \textit{(continuous reset gate)} \qquad (6)$$

$$\dot{\mathbf{h}} = (\mathbf{z}(t) - 1) \odot (\mathbf{h}(t) - \tanh(\mathbf{U}_h(\mathbf{r}(t) \odot \mathbf{h}(t)) + \mathbf{b}_h)) \qquad \textit{(continuous hidden state dynamics)} \qquad (7)$$

where $\dot{\mathbf{h}} \equiv \frac{\mathrm{d}\mathbf{h}(t)}{\mathrm{d}t}$. Since both $\sigma(\cdot)$ and $\tanh(\cdot)$ are smooth, this continuous limit is justified. The update gate $\mathbf{z}(t)$ again plays the role of a state-dependent time constant for memory decay. Furthermore, since $1 - \mathbf{z}(t) > 0$, it does not change the topological structure of the dynamics (only speed). For the following theoretical analysis sections (3 & 4), we can safely ignore the effects of $\mathbf{z}(t)$. A derivation of the continuous-time GRU can be found in appendix A.

## 3    STABILITY ANALYSIS OF A ONE DIMENSIONAL GRU

For a single GRU* ($d = 1$), (7) reduces to a one dimensional dynamical system where every variable is a scalar. The expressive power of a single GRU is quite limited, as only three stability structures (topologies) exist (see appendix B): (**A**) a single stable node, (**B**) a stable node and a half-stable node, and (**C**) two stable nodes separated by an unstable node (see Fig. 1). The corresponding temporal features are (A) decay to a fixed value, (B) decay to a fixed value, but from one direction halt at an intermediate value until perturbed, or (C) decay to one of two fixed values (bistability). The bistability can be used to capture a binary latent state in the sequence. It should be noted that a one dimensional continuous time autonomous system cannot exhibit oscillatory behavior, as is the case here (Hirsch et al., 2013).

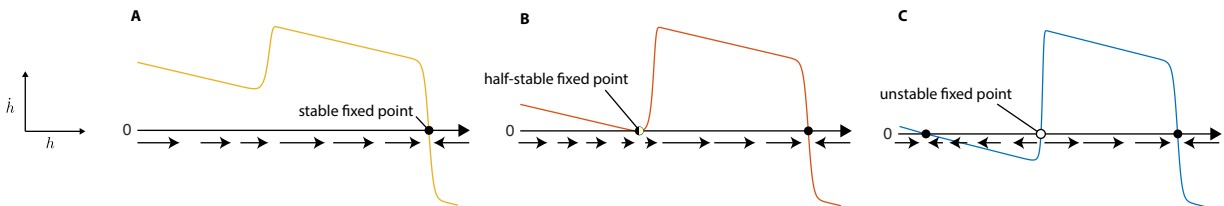

Figure 1: Three possible types of one dimensional flow for a single GRU. When $\dot{h} > 0$, $h(t)$ increases. This flow is indicated by a rightward arrow. Nodes ($\{h \mid \dot{h}(h) = 0\}$) are represented as circles and classified by their stability (Meiss, 2007).

The topology the GRU takes is determined by its parameters. If the GRU begins in a region of the parameter space corresponding to (A), we can smoothly vary the parameters to transverse (B) in the parameter space, and end up at (C).

---

*The number/dimension of GRUs referenced will indicate the dimension of the latent dynamics of a GRU network.

This is commonly known as a saddle-node bifurcation. Speaking generally, a bifurcation is the change in topology of a dynamical system, resulting from a smooth change in parameters. The point in the parameters space at which the bifurcation occurs is called the bifurcation point, and we will refer to the fixed point that changes its stability at the bifurcation point as the bifurcation fixed point. This corresponds to the parameters underlying (B) in our previous example. The codimension of a bifurcation is the number of parameters which must vary in order to achieve the bifurcation. In the case of our example, a saddle-node bifurcation is codimension-1 (Kuznetsov, 1998). Right before transitioning to (B), from (A), the flow near where the half-stable node would appear can exhibit arbitrarily slow flow. We will refer to these as *slow points* (Sussillo & Barak, 2012).

## 4    ANALYSIS OF A TWO DIMENSIONAL GRU

We will see that the addition of a second GRU opens up a substantial variety of possible topological structures when compared with the use of a single GRU. For notational simplicity, we denote the two dimensions of $\mathbf{h}$ as $x$ and $y$. We visualize the flow fields defined by (7) in 2-dimension as *phase portraits* which reveal the topological structures of interest. For starters, the phase portrait of two independent bistable GRUs can be visualized as Figure 2A. It clearly shows 4 stable states as expected, with a total of 9 stable fixed points. This could be thought of as a continuous-time continuous-space implementation of a finite state machine with 4 states (Fig. 2B). The 3 types of observed fixed points—stable (sinks), unstable (sources), and saddle points—exhibit locally linear dynamics, however, the global geometry is nonlinear and their topological structures can vary depending on their arrangement.

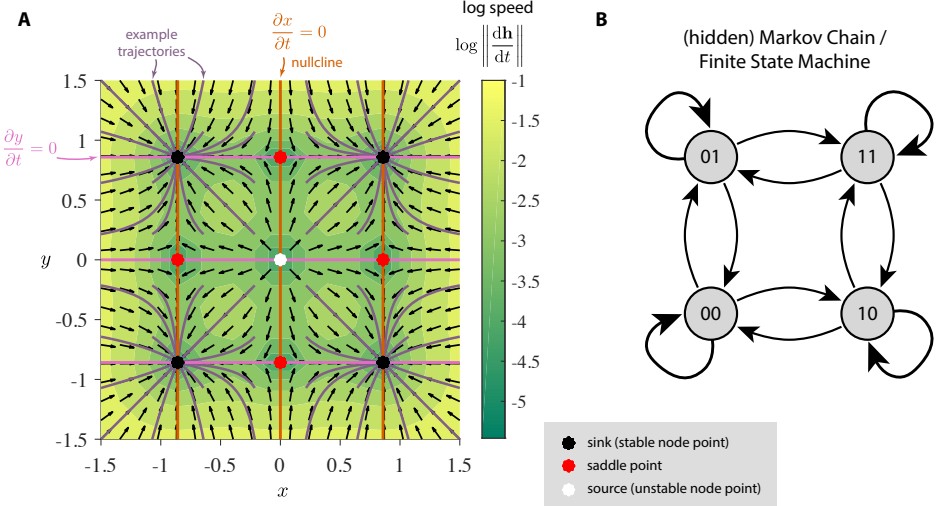

Figure 2: Illustrative example of two independent bistable GRUs. (**A**) Phase portrait. The flow field $\dot{\mathbf{h}} = [\dot{x}, \dot{y}]^\top$ is decomposed into direction (black arrows) and speed (color). Purple lines represent trajectories of the hidden state which converge to one of the four stable fixed points. Note the four quadrants coincide with the basin of attraction for each of the stable nodes. The fixed points appear when the x- and y-nullclines intersect. (**B**) The four stable nodes of this system can be interpreted as a continuous analogue of 4-discrete states with input-driven transitions.

We explored stability structures attainable by two GRUs. Due to the relatively large number of observed topologies, this section's main focus will be on demonstrating all observed local dynamical features obtainable by two GRUs. In addition, existence of two non-local dynamical features will be presented. A complete catalog of all observed topologies can be found in the appendix C, along with the parameters of every phase portrait depicted in this paper.

Before proceeding, let us take this time to describe all the local dynamical features observed. In addition to the previously mentioned three types of fixed points, two GRUs can exhibit a variety of bifurcation fixed points, resulting from regions of parameter space that separate all topologies restricted to simple fixed points (i.e stable, unstable, and saddle points). Behaviorally speaking, these fixed points act as hybrids between the previous three, resulting in a much richer set of obtainable dynamics. These bifurcation fixed points fall into two categories, separated by codimension. More specifically, two GRUs have been seen to feature both codimension-1 and codimension-2 bifurcation (fixed) points. Beginning with codimension-1, we have the saddle-node bifurcation fixed point, as expected from its existence in the single GRU case. We can further classify these points into two types. These can be thought of as both the fusion

of a stable fixed point and a saddle point, and the fusion of an unstable fixed point and a saddle point. We will refer to these fixed points as saddle-node bifurcation fixed points of the first kind and second kind respectively.

One type of codimension-2 bifurcation fixed point that has been observed in the two GRU system acts as the fusion of all three simple fixed points. More specifically, these points arise from the fusion of a stable fixed point, unstable fixed point, and two saddle points. All of these local structures are depicted in figure 3.

While the existence of simple fixed points was already demonstrated (see Fig. 2A). Figure 3A demonstrates the maximum number of fixed points observed in a two GRU system, for a given set of parameters. A closer look at this system reveals its potential interpretation as a continuous analogue of 5-discrete states with input-driven transitions, similar to that depicted in figure 2, implying additional GRUs are needed for any Markov process modeled in this manner, requiring more than five discrete states. We conjecture that the system depicted in figure 3A is the only eleven fixed point structure obtainable with two GRUs, as all observed structures containing the same number of fixed points are topologically equivalent to one another.

The addition of bifurcation fixed points opens the door to dynamically realize more sophisticated models. Take for example the four state system depicted in figure 3B. If the hidden state is set to initialize in the first quadrant of phase space, the trajectory will flow towards the codimension-2 bifurcation fixed point at the origin. Introducing noise through the input will stochastically cause the trajectory to approach the stable fixed point at (-1,-1) either directly, or by first flowing into one of the two saddle-node bifurcation fixed points of the first kind. Models of this sort can be used in a variety of applications, such as neural decision making (Wong & Wang (2006), Churchland & Cunningham (2014)).

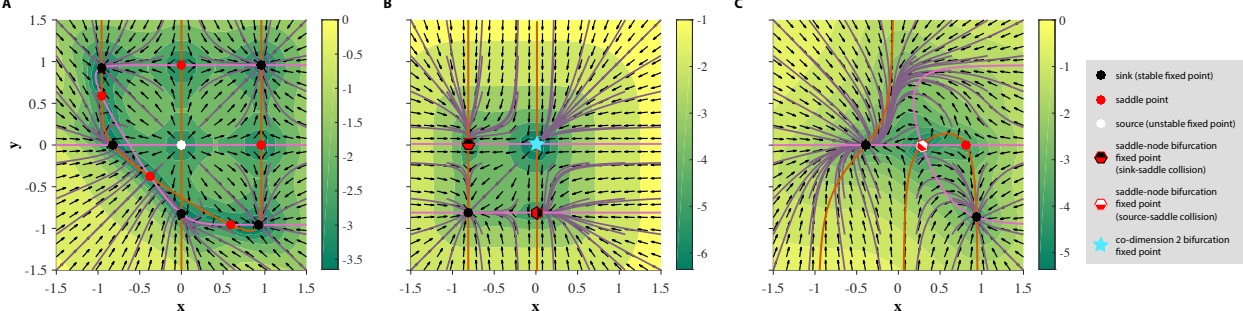

Figure 3: Existence of all observed simple fixed points and bifurcation fixed points with two GRUs, depicted in phase space. Orange and pink lines represent the x and y nullclines respectively. Purple lines indicate various trajectories of the hidden state. Direction of the flow is determined by the black arrows, where the colormap underlaying the figure depicts the magnitude of the velocity of the flow in log scale.

We will begin our investigation into the non-local dynamics observed with two GRUs by showing the existence of homoclinic orbits. A trajectory initialized on a homoclinic orbit will approach the same fixed point in both forward and backward time. We observe that two GRUs can exhibit one or two bounded planar regions of homoclinic orbits for a given set of parameters, as shown in figure 4A and 4B respectively. Any trajectory initialized in one of these regions will flow into the codimension-2 bifurcation fixed point at the origin, regardless of which direction time flows in. This featured behavior enables the accurate depiction of various models, including neuron spiking (Izhikevich, 2007).

In regards to the second non-local dynamic feature, it can be shown that two GRUs can exhibit an Andronov-Hopf bifurcation, whereby a stable fixed point bifurcates into an unstable fixed point surrounded by a limit cycle. Behaviorally speaking, a limit cycle is a type of attractor, in the sense that there exists a defined basin of attraction. However, unlike a stable fixed point, where trajectories initialized in the basin of attraction flow towards a single point, a limit cycle pulls trajectories into a stable periodic orbit around the unstable fixed point at its center. To demonstrate this phenomenon, let (8) define the parameters of (7).

$$U_z, U_r, b_z, b_r, b_h = 0, \ U_h = \frac{3}{2} \begin{bmatrix} \cos \alpha & -\sin \alpha \\ \sin \alpha & \cos \alpha \end{bmatrix} \tag{8}$$

where $\alpha \in \mathbb{R}^+$.

If $\alpha = \frac{\pi}{3}$, the system has a single stable fixed point (stable spiral), as depicted in figure 5A. If we continuously decrease $\alpha$, the system undergoes an Andronov-Hopf bifurcation approximately about $\alpha = \frac{\pi}{3.8}$. As $\alpha$ continuously decreases, the orbital period increases, and as the nullclines can be made arbitrarily close together, the length of this

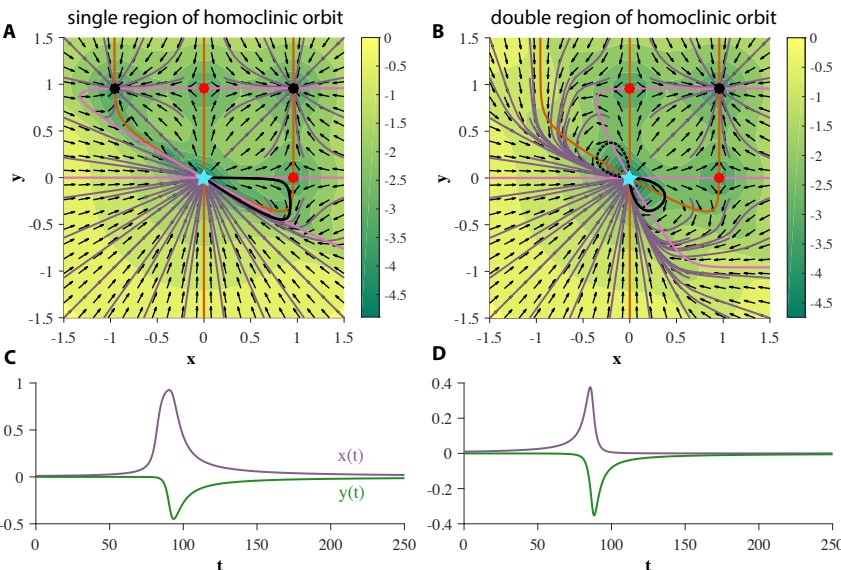

Figure 4: Two GRUs exhibit bounded regions of homoclinic orbits. 4C and 4D represent the hidden state as a function of time, for a single initial condition within the homoclinic region(s) of the single and double homoclinic region cases respectively (denoted by solid black trajectories in each corresponding phase portrait).

orbital period can be set arbitrarily, up to machine accuracy. Figure 5B shows an example of a relatively short orbital period, and figure 5C depicts the behavior seen for slower orbits.

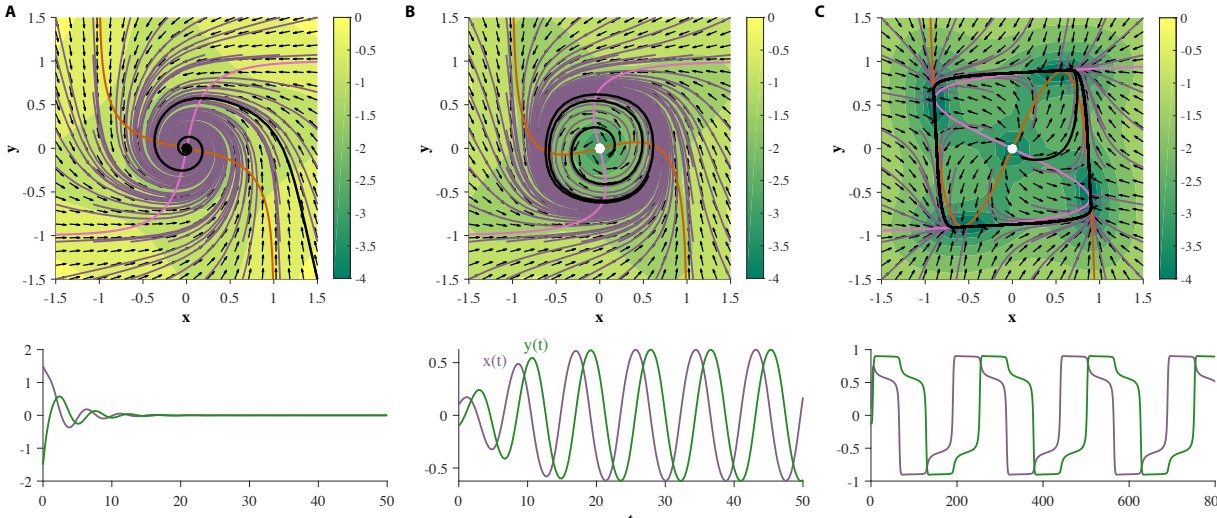

Figure 5: Two GRUs exhibit an Andronov-Hopf bifurcation, where the parameters are defined by (8). When $\alpha = \frac{\pi}{3}$ the system exhibits a single stable fixed point at the origin (Fig. 5A). If $\alpha$ decreases continuously, a limit cycle emerges around the fixed point, and the fixed point changes stability (Fig. 5B). Allowing $\alpha$ to decrease further increases the size and orbital period of the limit cycle (Fig. 5C). The bottom row represents the hidden state as a function of time, for a single trajectory (denoted by black trajectories in each corresponding phase portrait)

With finite-fixed point topologies and global structures out of the way, the next logical question to ask is, can two GRUs exhibit an infinite number of fixed points (countable or uncountable)? Such behavior is often desirable in models that require stationary attraction to non-point structures, such as line attractors and ring attractors. The short answer to this question is no.

**Lemma 1.** *Any two dimensional GRU can only have finitely many simple fixed points.*

This follows from Lefschetz theory (Guillemin & Pollack, 2010). The detailed proof can be found in appendix D, and is intended to give the reader intuition behind the result presented in the claim extended to aribitrary dimensional GRU in theorem 1.

**Theorem 1.** *Any finite dimensional GRU can only have finitely many simple fixed points and bifurcation fixed points.*

*Proof.* By definition of simple fixed points, the Jacobian of the dynamics at those fixed points have nonzero real parts, making them Lefschetz fixed points. Since GRU dynamics is asymptotically bounded on the compact set $[-1, 1]^d$, where $d$ is the number of GRUs, it follows from Lefshetz theory (Guillemin & Pollack, 2010) that there are finitely many simple fixed points. Furthermore, by construction, a bifurcation fixed point can only exist within a stability structure if and only if there exists a separate topology, such that the simple fixed points making up each bifurcation fixed point exist isolated from one another. Therefore, there can only exist finitely many isolated bifurcation fixed points. □

This eliminates the possibility of having countably many fixed points. Next, we show that there cannot be uncountably many non-isolated fixed points.

**Theorem 2.** *Any finite dimensional GRU cannot have a continuous attractor.*

*Proof.* We provide a sketch of a proof. Let $\mathbf{h}^* \in \mathbb{R}^d$ be a fixed point of a $d$-dimensional GRU, that is, $\mathbf{h}^* - \tanh(\mathbf{U}_h(\mathbf{r}(\mathbf{h}^*) \odot \mathbf{h}^*) + \mathbf{b}_h) = 0$. Now for any unit norm vector $\mathbf{k} \in \mathbb{S}^{d-1}$, and for any $\delta > 0$, we can show that there exist an $\epsilon > 0$ such that, $\|\dot{\mathbf{h}}(\mathbf{h}^* + \delta\mathbf{k}) - \dot{\mathbf{h}}(\mathbf{h}^*)\| = \|\dot{\mathbf{h}}(\mathbf{h}^* + \delta\mathbf{k})\| > \epsilon$. This can be argued by taking three cases into consideration, (a) $\mathbf{U}_r\mathbf{k} = 0$ and $\mathbf{U}_h(\sigma(\mathbf{b}_r) \odot \mathbf{k}) \neq 0$, (b) $\mathbf{U}_r\mathbf{k} = 0$ and $\mathbf{U}_h(\sigma(\mathbf{b}_r) \odot \mathbf{k}) = 0$, and (c) $\mathbf{U}_r\mathbf{k} \neq 0$. In each case, it reverts to a 1-dimensional problem where it can be trivially shown to have no continuous attractor around $\mathbf{h}^*$ along direction $\mathbf{k}$. Thus, we conclude that there is no continuous attractor in any direction. □

Despite this limitation, an approximation of a line attractor can be created using two GRUs. This approximation can be made arbitrarily close to an actual line attractor on a finite region in phase space, thereby satisfying computational needs on an arbitrary interval when scaled. We will refer to this phenomenon as a *pseudo-line attractor*. Figure 6 depicts an example of such an attractor.

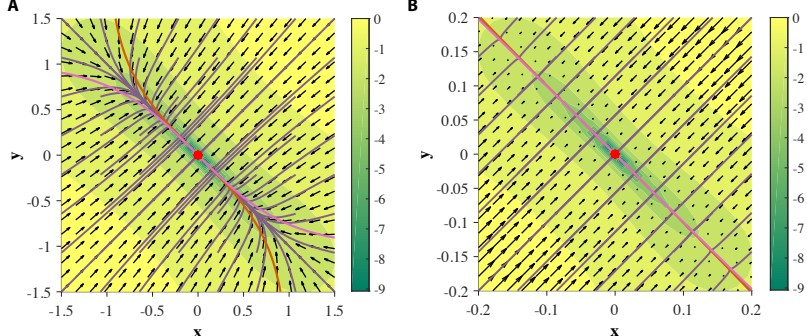

Figure 6: Two GRUs exhibit a pseudo-line attractor. Nullclines intersect at one point, but are close enough on a finite region to mimic an analytic line attractor in practice. 6A and 6B depict the phase portrait, on $[-1.5, 1.5]^2$ and $[-0.2, 0.2]^2$ respectively.

We conclude this section with a discussion of slow points in the two GRU system. As a logical extension to the single GRU system, slow points occur where the nullclines are sufficiently close together, but do not intersect, as demonstrated in figure 7. Given the previously discussed classes of dynamic behavior for two GRUs, slow points can only exist so long as the *potential* for a saddle-node bifurcation fixed point is possible in the location of the desired slow point, given an appropriate change in parameters, as they result from the collision and annihilation of two fixed points. This observation is consistent with the single GRU case, as slow points can only exist for a single fixed point. This would imply that given the one fixed point case, a maximum of five slow points are possible. However, this would imply that there must exist a six fixed point case by which five of the six fixed points exist at saddle-node bifurcation fixed points, which has not been observed (see appendix C). Despite this shortcoming, four simultaneous slow points are obtainable, as shown in figure 5C.

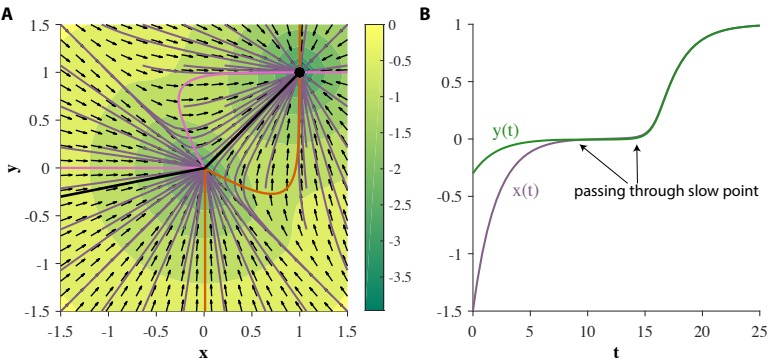

Figure 7: An example of a slow point about the origin, obtainable with two GRUs. Initial conditions satisfying $y < -x$ are attracted to the slow point at the origin before a secondary attraction to the sink. 7A depicts the phase portrait of the system, and 7B shows the dynamics of the hidden state for a single initial condition (denoted by a black trajectory on 7A).

## 5 NUMERICAL EXPERIMENTS

As a means to put our theory to practice, in this section we explore several examples of time series prediction of continuous time planar dynamical systems using two GRUs. Results from the previous section indicate what dynamical features can be learned by this RNN, and suggest cases by which training will fail. All of the following computer experiments consist of an RNN, by which the hidden layer is made up of two GRUs, followed by a linear output layer. The network is trained to make a 29-step prediction from a given initial observation, and no further input through prediction. As such, to produce accurate predictions, the RNN must rely solely on the hidden layer dynamics.

We train the network to minimize the following multi-step loss function:

$$\mathcal{L}(\theta) = \frac{1}{T} \sum_{i=1}^{N_{\text{traj}}} \sum_{k=1}^{T} \|\hat{\mathbf{w}}_i(k; \mathbf{w}_i(0)) - \mathbf{w}_i(k)\|_2^2 \tag{9}$$

where $\theta$ are the parameters of the GRU and linear readout, $T = 29$ is the prediction horizon, $\mathbf{w}_i(t)$ is the $i$-th time series generated by the true system, and $\hat{\mathbf{w}}(k; \mathbf{w}_0)$ is $k$-step the prediction given $\mathbf{w}_0$.

The hidden states are initialized at zero for each trajectory. The RNN is then trained for 4000 epochs, using ADAM (Kingma & Ba, 2014) in whole batch mode to minimize the loss function, i.e., the mean square error between the predicted trajectory and the data. $N_{\text{traj}} = 667$ time series were used for training. Figure 8 depicts the experimental results of the RNN's attempt at learning each dynamical system we describe below.

### 5.1 LIMIT CYCLE

To test if two GRUs can learn a limit cycle, we use a simple nonlinear oscillator called the FitzHugh-Nagumo Model. The FitzHugh-Nagumo model is defined by:

$$\dot{x} = x - \frac{x^3}{3} - y + I_{\text{ext}}, \quad \tau \dot{y} = x + a - by \tag{10}$$

where in this experiment we will chose $\tau = 12.5$, $a = 0.7$, $b = 0.8$, and $I_{\text{ext}} = \mathcal{N}(0.7, 0.04)$. Under this choice of model parameters, the system will exhibit an unstable fixed point (unstable spiral) surrounded by a limit cycle (Fig. 8). As shown in section 4, two GRUs are capable of representing this topology. The results of this experiment verify this claim (Fig. 8), as two GRUs can capture topologically equivalent dynamics.

### 5.2 LINE ATTRACTOR

As discussed in section 4, two GRUs can exhibit a pseudo-line attractor, by which the system mimics an analytic line attractor. We will use the simplest representation of a planar line attractor:

$$\dot{x} = -x, \quad \dot{y} = 0 \tag{11}$$

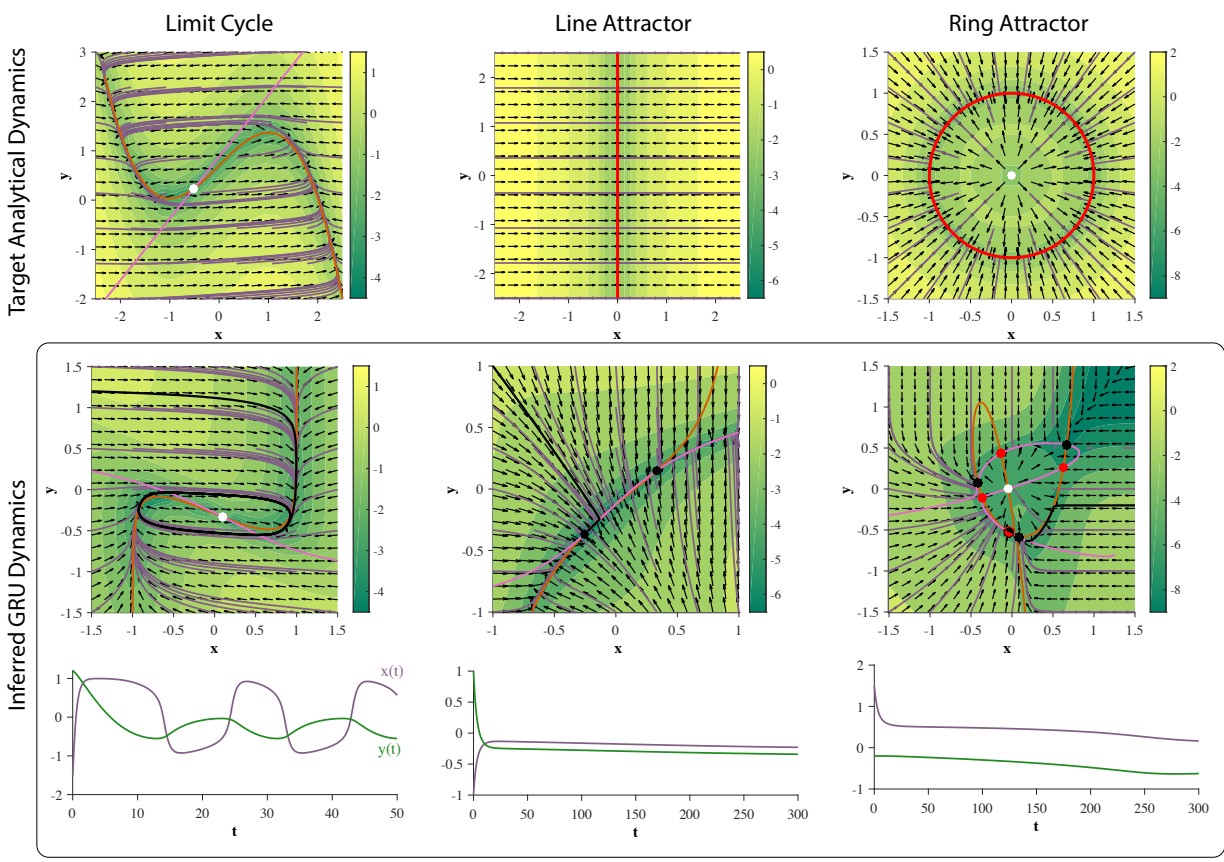

Figure 8: **Training 2-dim GRUs.** (top row) Phase portraits of target dynamical systems. Red solid line represents 1-dimensional attractor. See main text for each system. (middle row) GRU dynamics learned from corresponding 29-step forecasting tasks. Note that the prediction is an affine transformation of the hidden state. (bottom row) An example time series generated through closed-loop prediction of the trained GRU (denoted by a black trajectory). Note that GRU fails to learn the ring attractor.

This system will exhibit a line attractor along the $y$-axis, at $x = 0$ (Fig. 8). Trajectories will flow directly perpendicular towards the attractor. We added white Gaussian noise $\mathcal{N}(0, 0.1I)$ in the training data. While the hidden state dynamics of the trained network do not perfectly match that of an analytic line attractor, there exists a subinterval near each of the fixed points acting as a pseudo-line attractor (Fig. 8). As such, the added affine transformation (linear readout) can scale and reorient this subinterval as is required by a given problem, thereby mimicking a line attractor.

### 5.3 RING ATTRACTOR

For this experiment, a dynamical system representing a standard ring attractor of radius one is used:

$$\dot{x} = -(x^2 + y^2 - 1)x, \quad \dot{y} = -(x^2 + y^2 - 1)y \tag{12}$$

This system exhibits an attracting ring, centered around an unstable fixed point. We added Gaussian noise $\mathcal{N}(0, 0.1I)$ to the training data.

Two GRUs will not be able to accurately capture the system's continuous attractor dynamics as expected from theorem 3. The results of this experiment are demonstrated in figure 8. As expected, the RNN fails to capture the proper dynamical features of the ring attractor. Rather, the hidden state dynamics fall into an observed finite fixed point topology (see case xxix in appendix C). In addition, we robustly see this over multiple initializations, and the quality of approximation improves as the dimensionality of GRU increases (Fig. 9).

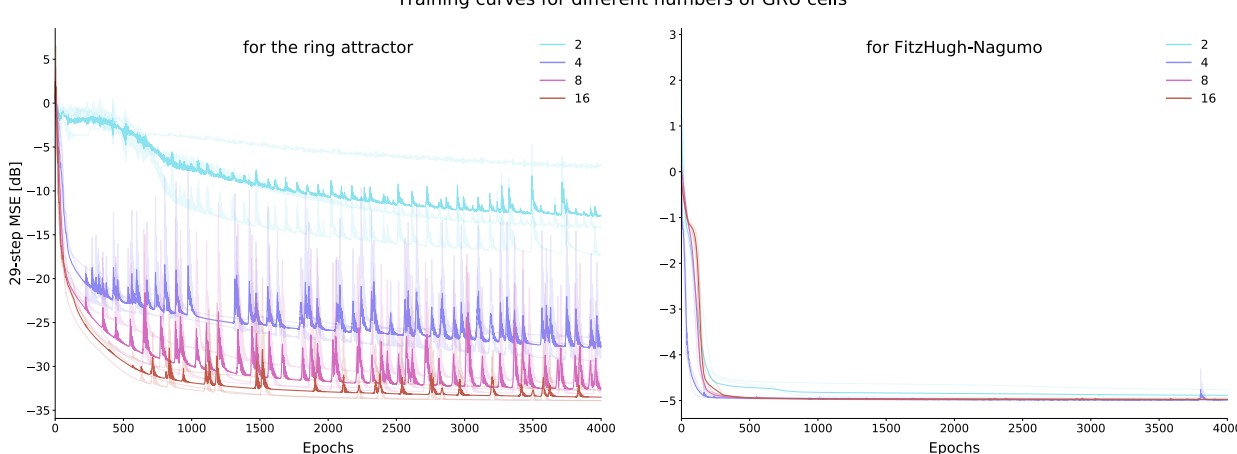

Figure 9: Learning curve (training loss) for ring attractor (left) and the FitzHugh-Nagumo (right) dynamics. Note that the performance of the ring attractor improves as the dimensinoality of the GRU increases unlike the FHN dynamics. Four network sizes (2, 4, 8, 16 dimensional GRU) were trained 3 timeswith different initializations. Initial learning rate for ADAM was tuned using Bayesian optimization procedure.

## 6 CONCLUSION

Our analysis shows the rich but limited classes of dynamics the GRU can approximate in one, two, and arbitrary dimensions. We developed a new theoretical framework to analyze GRUs as a continuous dynamical system, and showed that two GRUs can exhibit a variety of expressive dynamic features, such as limit cycles, homoclinic orbits, and a substantial catalog of stability structures and bifurcations. However, we also showed that finitely many GRUs cannot exhibit the dynamics of an arbitrary continuous attractor. These claims were then experimentally verified in two dimensions. We believe these findings also unlock new avenues of research on the trainability of recurrent neural networks. Although we have analyzed GRUs only in 1- and 2- dimensions in near exhaustive, we believe that the insights extends to higher-dimensions. We leave rigorous analysis of higher-dimensional GRUs as future work.

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

## A  CONTINUOUS TIME SYSTEM DERIVATION

We begin with the fully gated GRU as a discrete time system, where the input vector $x_t$ has been set equal to zero, as depicted in (13) - (15), where $\odot$ is the Hadamard product, and $\sigma$ is the *sigmoid function*.

$$z_t = \sigma(U_z h_{t-1} + b_z) \tag{13}$$
$$r_t = \sigma(U_r h_{t-1} + b_r) \tag{14}$$
$$h_t = z_t \odot h_{t-1} + (1 - z_t) \odot \tanh\left(U_h(r_t \odot h_{t-1}) + b_h\right) \tag{15}$$

We recognize that (15) is a forward Euler discretization of a continuous time dynamical system. This allows us to consider the underlying continuous time dynamics on the basis of the discretization. The following steps are a walk through of the derivation:

Since $z_t$ is a bounded function on $\mathbb{R}\ \forall t$, there exists a function $\tilde{z}_t$, such that $z_t + \tilde{z}_t = 1$ at each time step (due to the symmetry of $z_t$, $\tilde{z}_t$ is the result of vertically flipping $z_t$ about 0.5, the midpoint of its range). As such, we can rewrite (15) with $\tilde{z}_t$ as depicted in (16).

$$h_t = (1 - \tilde{z}_t) \odot h_{t-1} + \tilde{z}_t \odot \tanh\left(U_h(r_t \odot h_{t-1}) + b_h\right) \tag{16}$$

where,

$$\tilde{z}_t = \sigma(\tilde{U}_z h_{t-1} + \tilde{b}_z) \tag{17}$$
$$h_t = h_{t-1} - \tilde{z}_t \odot h_{t-1} + \tilde{z}_t \odot \tanh\left(U_h(r_t \odot h_{t-1}) + b_h\right) \tag{18}$$
$$h_t - h_{t-1} = -\tilde{z}_t \odot \left(h_{t-1} - \tanh\left(U_h(r_t \odot h_{t-1}) + b_h\right)\right) \tag{19}$$

Let $h(t) \equiv h_{t-1}$. As a result, we can say $\tilde{z}_t \equiv \tilde{z}(t)$ and $r_t \equiv r(t)$, as depicted in (20).

$$h(t+1) - h(t) = -\tilde{z}(t) \odot \left(h(t) - \tanh\left(U_h(r(t) \odot h(t)) + b_h\right)\right) \tag{20}$$

where,

$$\tilde{z}(t) = \sigma(\tilde{U}_z h(t) + \tilde{b}_z) \tag{21}$$
$$r(t) = \sigma(U_r h(t) + b_r) \tag{22}$$

Let $\Delta t$ define an arbitrary time interval. Then (20) becomes,

$$h(t + \Delta t) - h(t) = -\tilde{z}(t) \odot \left(h(t) - \tanh U_h(r(t) \odot h(t)) + b_h\right)\Delta t \tag{23}$$

Dividing both sides of the equation by $\Delta t$ yields (24).

$$\frac{h(t + \Delta t) - h(t)}{\Delta t} = -\tilde{z}(t) \odot \left(h(t) - \tanh U_h(r(t) \odot h(t)) + b_h\right) \tag{24}$$

If we take the limit as $\Delta t \to 0$, we get the analogous continuous time system to (13) - (15),

$$\dot{h} = -\tilde{z}(t) \odot \left(h(t) - \tanh\left(U_h(r(t) \odot h(t)) + b_h\right)\right) \tag{25}$$

where $\dot{h} \equiv \frac{dh(t)}{dt}$

Finally, we can rewrite (25) as follows:

$$\dot{h} = (z(t) - 1) \odot \left(h(t) - \tanh\left(U_h(r(t) \odot h(t)) + b_h\right)\right) \tag{26}$$

where

$$z(t) = \sigma(U_z h(t) + b_z) \tag{27}$$

## B  SINGLE GRU FIXED POINT PROOFS

The fixed points of our continuous time system (25) exist where the derivative $\dot{h} = 0$. In the single GRU case, the Hadamard product reduces to standard scalar multiplication, yielding,

$$0 = (z(t) - 1)[h^* - \tanh\left(U_h r(t) h^* + b_h\right)] \tag{28}$$

where $z(t)$ and $r(t)$ are defined by (27) and (22) respectively, and $h^* \in \mathbb{R}$ represents a solution of (28).

We can divide out $z(t) - 1$, indicating that the update gate does not play a part in the stability of the system. For simplicity, lets expand $r(t)$ in (28) by its definition (22).

$$0 = \tanh\left(U_h \sigma(U_r h^* + b_r) h^* + b_h\right) - h^* \tag{29}$$

where $U_h, b_h, U_r, b_r \in \mathbb{R}$.

**Lemma 2.** *for all $U_h, b_h, U_r, b_r$, there exists $h^*$ such that (29) is satisfied.*

*Proof.* The hyperbolic tangent function is continuous and bounded on $\mathbb{R}$, having a range of $(-1, 1)$. $-h(t) \equiv -h$ is monotonic and achieves all values on $\mathbb{R}$, as $-h$ is bijective on $\mathbb{R}$. Thus, their sum is unbounded and obtains every value on $\mathbb{R}$ at least once. By the intermediate value theorem, there is at least one point $h^*$ such that (29) is satisfied, regardless of the choice of parameters $U_h, b_h, U_r, b_r$. □

**Lemma 3.** *There exists a set of parameters $U_h, b_h, U_r, b_r$ such that there exists one, two, or three solutions to (29).*

*Proof.* We will prove this lemma by showing the existence of each case. Let $U_r = 80$, $b_r = 40$, and $U_h = -60$. We then allow $b_h$ to vary. The existence of each of the three cases are shown in figure 1.

If $b_h = -1$, there exists a single solution to (29). If $b_h$ decreases continuously, a second root appears and splits in two. Analogously, the system (25) goes through a *saddle-node bifurcation*, where a half-stable fixed point appears, and splits into a stable/unstable fixed point pair. □

**Theorem 3.** *For any choice of parameters $U_r, b_r, U_h, b_h$, there can only exist one, two, or three solutions to (29), and all solutions exist on the interval (-1,1).*

*Proof.* We begin with the argument of the hyperbolic tangent function in (29),

$$U_h \sigma(U_r h + b_r) h + b_h \tag{30}$$

Taking the derivative of (30) yields,

$$\frac{U_h(1 + e^{-U_r h - b_r}) + U_h U_r h e^{-U_r h - b_r}}{(1 + e^{-U_r h - b_r})^2} \tag{31}$$

Setting (31) to zero and simplifying will allow us to find the nontrivial critical points of (30), as shown in (32). Note that if $U_h = 0$, (30) is equal to $b_h \; \forall h$, yielding no critical points.

$$1 + e^{-U_r h - b_r}(1 + U_r h) = 0 \tag{32}$$

Let $x \equiv e^{-b_r}$ and $\hat{h} \equiv U_r h$, and solve (32) for $\hat{h}$.

$$\hat{h} = -W(\frac{1}{xe}) - 1 \tag{33}$$

where W is principal branch of the *Lambert W function*. Therefore, (30) has exactly one local maximum or minimum, so long as $U_h \neq 0$.

Now consider,

$$\tanh\left(U_h \sigma(U_r h + b_r) h + b_h\right) \tag{34}$$

The hyperbolic tangent function preserves intervals of monotonic behavior in its argument. Therefore, (34) has at most one local maximum or minimum.

We take into account the fact that the hyperbolic tangent function bounds its argument on the interval (-1,1). If there exists a subset $S = [a, 1]$, for some $a \in [-1, 1)$ such that (34) is increasing, then there exists a $k \in S$ such that when $h = k$ (35) and (36) hold.

$$\frac{d}{dh}\left(\tanh\left(U_h \sigma(U_r h + b_r) h + b_h\right) - h\right) = 0 \tag{35}$$

$$\frac{d}{dh}\left(\tanh\left(U_h \sigma(U_r h + b_r) h + b_h\right) - h\right) < 0, \forall h > k \tag{36}$$

This result in conjunction with the previous two lemmas completes the proof. □

## C  ALL TOPOLOGICAL STABILITY STRUCTURES OBSERVED WITH TWO GRUS

Table 2 depicts all observed topologies of multiple-fixed point structures using two GRUs. Figure 10 displays an example of a phase portrait from a two GRU system for each case listed in 2. Note that all fixed points are denoted by a red dot, regardless of classification. Table 3 lists the parameters used for each of the observed cases. Note that all the update gate parameters are set to zero.

Each case in this paper was discovered by hand by considering the geometric constraints on the structure of nullclines for both the decoupled and coupled system (i.e reset gate inactive and active respectively). An exhaustive analysis on the one dimensional GRU allowed for a natural extension into the two dimensional decoupled GRU. Upon establishing a set of base cases (a combinatorial argument regarding all possible ways the decoupled nullclines [topologically conjugate to linear and cubic polynomials] can intersect) From these base cases, the reset gate can be used as a means of bending and manipulating structure of the decoupled nullclines in order to obtain new intersection patterns in the coupled system.

Table 1: Parameters Used for all Phase Portraits in Section 4

| Figure | Uh11 | Uh12 | Uh21 | Uh22 | Ur11 | Ur12 | Ur21 | Ur22 | bh1 | bh2 | br1 | br2 |
|--------|------|------|------|------|------|------|------|------|-----|-----|-----|-----|
| 2 | 3 | 0 | 0 | 3 | 0 | 0 | 0 | 0 | 0 | 0 | 0 | 0 |
| 3a | 2 | 0 | 0 | 2 | 5 | 8 | 8 | 5 | 0 | 0 | 5 | 5 |
| 3b | 2 | 0 | 0 | 2 | -1 | 0 | 0 | -1 | 0 | 0 | 0 | 0 |
| 3c | 2 | 0 | 0 | 2 | 1 | -2 | 3 | 1 | -0.06 | 0 | 0.2 | -0.85 |
| 4a | 2 | 0 | 0 | 2 | 5 | 9 | 5 | 9 | 0 | 0 | 0 | 0 |
| 4b | 2 | 0 | 0 | 2 | 5 | 9 | 9 | 5 | 0 | 0 | 0 | 0 |
| 5a | 1.5 | -2.598 | 2.598 | 1.5 | 0 | 0 | 0 | 0 | 0 | 0 | 0 | 0 |
| 5b | 2.4271 | -1.7634 | 1.7634 | 2.4271 | 0 | 0 | 0 | 0 | 0 | 0 | 0 | 0 |
| 5c | 2.9665 | -0.4471 | 0.4471 | 2.9665 | 0 | 0 | 0 | 0 | 0 | 0 | 0 | 0 |
| 6a | 0.1 | -0.1 | -1 | 0 | 0 | 0 | 0 | 0 | 0 | 0 | 0 | 0 |
| 7a | 2 | 2 | 2 | 2 | 100 | 100 | 100 | 100 | 0.01 | 0 | 0 | 0 |

Table 2: Multiple Fixed Point Stability Structures Obtainable with two GRUs

| Case | Fixed Points | Sinks | Sources | Saddle Points | Saddle Point and Stable Node Collisions | Saddle Point and Unstable Node Collisions | Codim. 2 Bifurcation Point | Figure Reference |
|---|---|---|---|---|---|---|---|---|
| i | 2 | 1 | - | - | 1 | - | - | 10i |
| ii | 3 | 2 | - | 1 | - | - | - | 10ii |
| iii | 3 | 1 | - | - | 2 | - | - | 10iii |
| iv | 4 | 1 | - | - | 2 | - | 1 | 10iv |
| v | 4 | 2 | - | 1 | - | - | 1 | 10v |
| vi | 4 | 2 | - | 1 | - | 1 | - | 10vi |
| vii | 4 | 2 | - | 1 | 1 | - | - | 10vii |
| viii | 4 | 1 | - | - | 3 | - | - | 10viii |
| ix | 5 | 2 | 1 | 2 | - | - | - | 10ix |
| x | 5 | 3 | - | 2 | - | - | - | 10x |
| xi | 5 | 3 | - | 1 | - | 1 | - | 10xi |
| xii | 5 | 2 | - | 1 | - | 2 | - | 10xii |
| xiii | 5 | 2 | - | 1 | 1 | 1 | - | 10xiii |
| xiv | 5 | - | 1 | - | 4 | - | - | 10xiv |
| xv | 6 | 2 | - | 1 | 2 | 1 | - | 10xv |
| xvi | 6 | 3 | - | 2 | - | 1 | - | 10xvi |
| xvii | 6 | 2 | 1 | 2 | 1 | - | - | 10xvii |
| xviii | 6 | 3 | - | 2 | - | - | 1 | 10xviii |
| xix | 6 | 2 | 1 | 2 | - | 1 | - | 10xix |
| xx | 6 | 3 | - | 2 | 1 | - | - | 10xx |
| xxi | 6 | 1 | 1 | 1 | 3 | - | - | 10xxi |
| xxii | 7 | 3 | 1 | 3 | - | - | - | 10xxii |
| xxiii | 7 | 2 | 2 | 3 | - | - | - | 10xxiii |
| xxiv | 7 | 4 | 3 | - | - | - | - | 10xxiv |
| xxv | 7 | 2 | 1 | 2 | 2 | - | - | 10xxv |
| xxvi | 7 | 3 | - | 2 | 2 | - | - | 10xxvi |
| xxvii | 7 | 3 | - | 2 | 1 | 1 | - | 10xxvii |
| xxviii | 8 | 4 | - | 3 | - | 1 | - | 10xxviii |
| xxix | 8 | 3 | 1 | 3 | 1 | - | - | 10xxix |
| xxx | 8 | 3 | - | 2 | 2 | 1 | - | 10xxx |
| xxxi | 9 | 4 | 1 | 4 | - | - | - | 10xxxi |
| xxxii | 9 | 3 | 1 | 3 | 2 | - | - | 10xxxii |
| xxxiii | 9 | 5 | - | 4 | - | - | - | 10xxxiii |
| xxxiv | 10 | 4 | 1 | 4 | 1 | - | - | 10xxxiv |
| xxxv | 10 | 5 | - | 4 | - | 1 | - | 10xxxv |
| xxxvi | 11 | 5 | 1 | 5 | - | - | - | 10xxxvi |

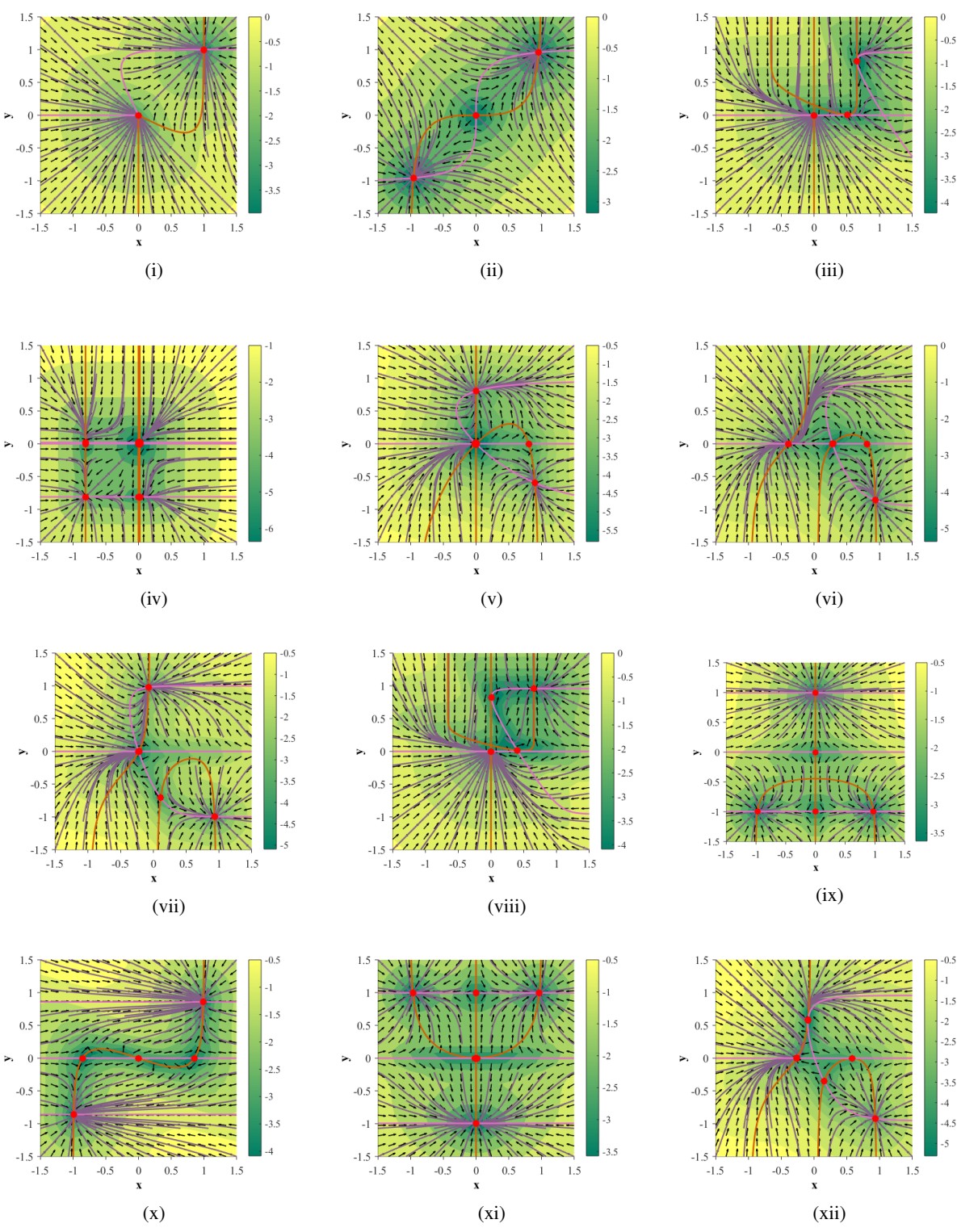

Figure 10: Thirty six multiple fixed-point topologies obtainable with two GRUs, depicted in phase space. Orange and pink lines represent the x and y nullclines respectively. Red dots indicate fixed points. Each subfigure contains 64 purple lines, indicating trajectories in forward time, whose initial conditions were chosen to be evenly spaced on the vertices of a square grid on $[-1.5, 1.5]^2$. Direction of the flow is determined by the black arrows, and the underlaying color map represents the magnitude of the velocity of the flow in log scale.

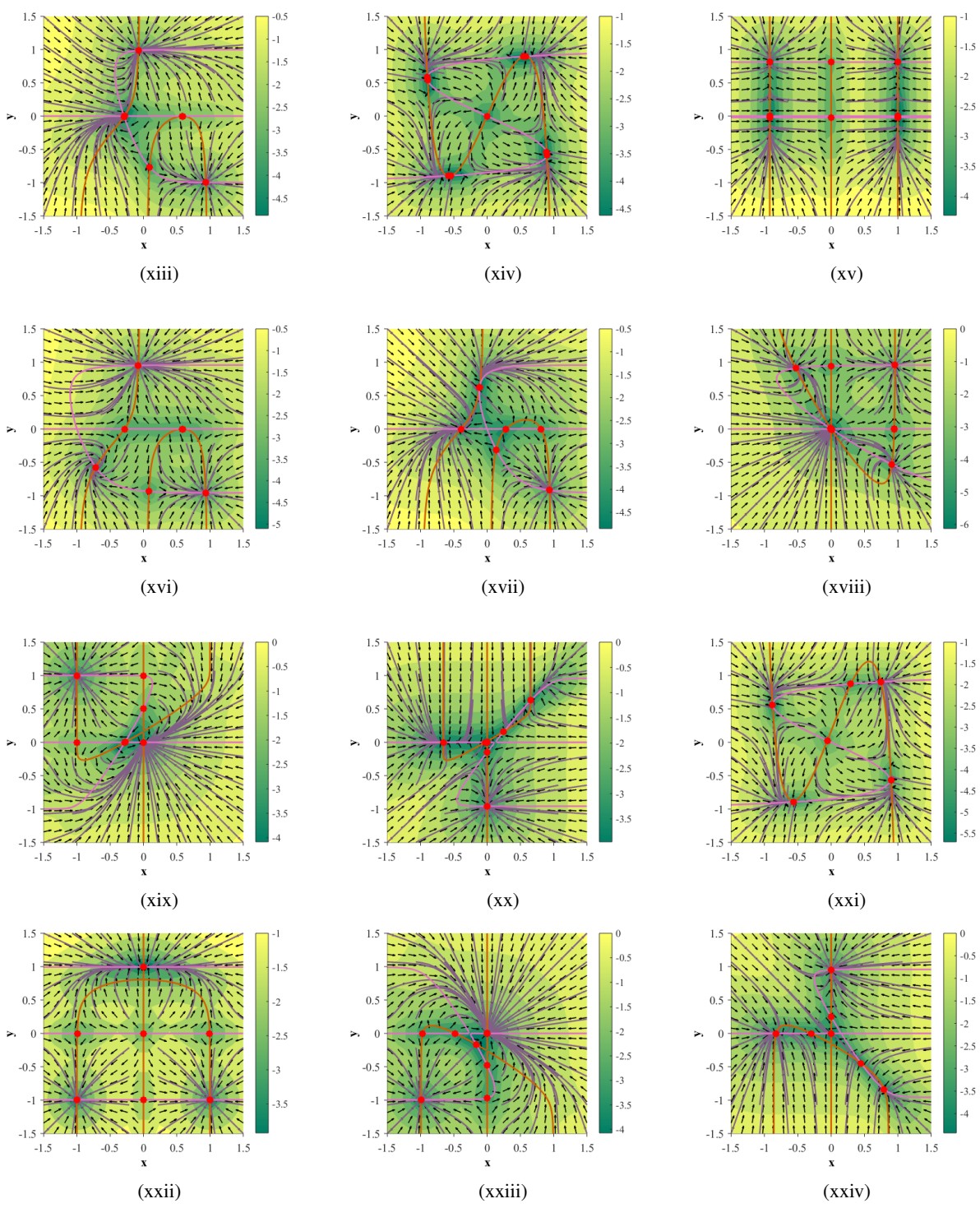

Figure 10: Thirty six multiple fixed-point topologies obtainable with two GRUs, depicted in phase space. Orange and pink lines represent the x and y nullclines respectively. Red dots indicate fixed points. Each subfigure contains 64 purple lines, indicating trajectories in forward time, whose initial conditions were chosen to be evenly spaced on the vertices of a square grid on $[-1.5, 1.5]^2$. Direction of the flow is determined by the black arrows, and the underlaying color map represents the magnitude of the velocity of the flow in log scale.

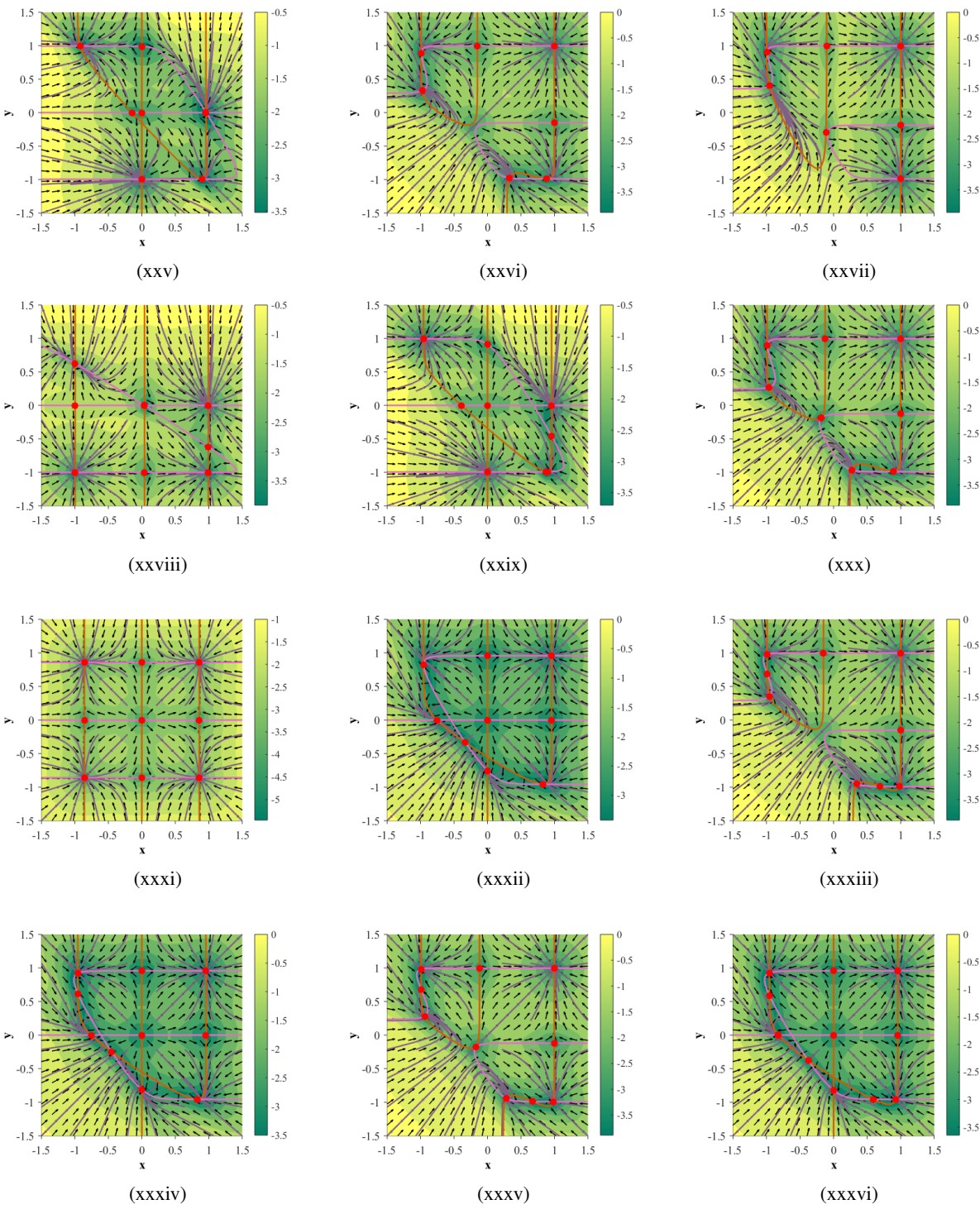

Figure 10: Thirty six multiple fixed-point topologies obtainable with two GRUs, depicted in phase space. Orange and pink lines represent the x and y nullclines respectively. Red dots indicate fixed points. Each subfigure contains 64 purple lines, indicating trajectories in forward time, whose initial conditions were chosen to be evenly spaced on the vertices of a square grid on $[-1.5, 1.5]^2$. Direction of the flow is determined by the black arrows, and the underlaying color map represents the magnitude of the velocity of the flow in log scale.

Table 3: Parameters of each multiple fixed-point stability structure example

| Case | Uh11 | Uh12 | Uh21 | Uh22 | Ur11 | Ur12 | Ur21 | Ur22 | bh1 | bh2 | br1 | br2 |
|------|------|------|------|------|------|------|------|------|-----|-----|-----|-----|
| i | 2 | 2 | 2 | 2 | 100 | 100 | 100 | 100 | 0 | 0 | 0 | 0 |
| ii | 2 | 2 | 2 | 2 | 0 | 0 | 0 | 0 | 0 | 0 | 0 | 0 |
| iii | 1.2 | 0 | 0 | 2 | 5 | 12 | 8 | 5 | 0 | 0 | -0.22 | -0.5 |
| iv | 2 | 0 | 0 | 2 | -1 | 0 | 0 | -1 | 0 | 0 | 0 | 0 |
| v | 2 | 0 | 0 | 2 | 1 | -1 | 1 | 1 | 0 | 0 | 0 | 0 |
| vi | 2 | 0 | 0 | 2 | 1 | -2 | 3 | 1 | -0.06 | 0 | 0.2 | -0.85 |
| vii | 2 | 0 | 0 | 4 | 1 | -2 | 3 | 1 | -0.06 | 0 | -0.3 | -0.42 |
| viii | 1.2 | 0 | 0 | 2 | 5 | 20.5 | 8 | 5 | 0 | 0 | -0.275 | -3.3 |
| ix | 3 | 3 | 0 | 3 | 0 | 0 | 0 | 0 | 0 | 0 | 0 | 0 |
| x | 6 | 0 | 0 | 6 | 0 | -2 | 0 | 0 | 0 | 0 | -1.695 | 0 |
| xi | 6 | 0 | 0 | 6 | 0 | 1 | 0 | 0 | 0 | 0 | -2.5 | 0 |
| xii | 2 | 0 | 0 | 2 | 1 | -2 | 3 | 1 | -0.055 | 0 | -0.1 | -0.02 |
| xiii | 2 | 0 | 0 | 4 | 1 | -2 | 3 | 1 | -0.06 | 0 | -0.085 | -0.22 |
| xiv | 2.9674 | -0.4409 | 0.4409 | 2.9674 | 0 | 0 | 0 | 0 | 0 | 0 | 0 | 0 |
| xv | 6 | 0 | 0 | 2 | -1 | 0 | 0 | 0 | 0 | 0 | 0 | 0 |
| xvi | 2 | 0 | 0 | 2 | 1 | -2 | 3 | 1 | -0.06 | 0 | -0.08 | 3 |
| xvii | 2 | 0 | 0 | 2 | 1 | -2 | 3 | 1.1 | -0.06 | 0 | 0.2 | 0 |
| xviii | 2 | 0 | 0 | 2 | 3 | 2 | 2 | 3 | 0 | 0 | 0 | 0 |
| xix | 10 | 0 | 0 | 6 | -3 | 5 | -5 | 3 | 0 | 0 | -3 | -3 |
| xx | 1.2 | 0 | 0 | 2 | -17 | 35 | 10 | -8 | 0 | 0 | 1.2 | -1.2 |
| xxi | 2.9763 | -0.376 | 0.376 | 2.9763 | 0 | 0 | 0 | 0 | 0.0315 | 0 | -0.015 | -0.015 |
| xxii | 6 | 0 | 0 | 6 | 0 | -2 | 0 | 0 | 0 | 0 | 0 | 0 |
| xxiii0 | 3 | 0 | 0 | 3 | -3 | -5 | -5 | -3 | 0 | 0 | -2 | -2 |
| xxiv | 1.5 | 0 | 0 | 2 | -4 | -7 | 8 | 5 | 0 | 0 | -0.4 | -1.2 |
| xxv | 2 | 0 | 0 | 3 | 12.4 | 11.6 | -8 | -5 | 0 | 0 | 1.8 | 7 |
| xxvi | 3 | 0 | 0 | 3 | 5.175 | 9 | 9 | 5.175 | 0.3 | 0.3 | 3.95 | 3.95 |
| xxvii | 7 | 0 | 0 | 3 | 6 | 3 | 9 | 6 | 0.62 | 0.373 | 4 | 3.4 |
| xxviii | 6 | 0 | 0 | 10 | 0 | 0 | -5 | -8 | -0.08 | 0 | 0 | -2 |
| xxix | 2 | 0 | 0 | 3 | -10 | -11.6 | 8 | 5 | 0 | 0 | 4 | 4.8 |
| xxx | 3 | 0 | 0 | 3 | 5.26 | 9 | 9 | 5.26 | 0.25 | 0.25 | 3.95 | 3.95 |
| xxxi | 3 | 0 | 0 | 3 | 0 | 0 | 0 | 0 | 0 | 0 | 0 | 0 |
| xxxii | 2 | 0 | 0 | 2 | 5 | 8 | 8 | 5 | 0 | 0 | 4.4 | 4.4 |
| xxxiii | 3 | 0 | 0 | 3 | 6 | 9 | 9 | 6 | 0.3 | 0.3 | 3.75 | 3.75 |
| xxxiv | 1 | 0 | 0 | 1 | 5 | 8 | 8 | 5 | 0 | 0 | 4.4 | 4.9 |
| xxxv | 3 | 0 | 0 | 3 | 6 | 9 | 9 | 6 | 0.24 | 0.24 | 3.95 | 3.95 |
| xxxvi | 2 | 0 | 0 | 2 | 5 | 8 | 8 | 5 | 0 | 0 | 5 | 5 |

# D   PROOF OF LEMMA 1

We begin this proof by showing that all fixed points obtainable with two GRUs are Lefschetz fixed points. To show this is the case let (37) expand our previous notation. We set all elements in $U_z$ and $b_z$ to zero, as the update gate plays no part in the topology of (7) (shown in appendix B).

$$U_h = \begin{bmatrix} U_{h11} & U_{h12} \\ U_{h21} & U_{h22} \end{bmatrix}, U_r = \begin{bmatrix} U_{r11} & U_{r12} \\ U_{r21} & U_{r22} \end{bmatrix}, b_h = \begin{bmatrix} b_{h1} \\ b_{h2} \end{bmatrix}, b_r = \begin{bmatrix} b_{r1} \\ b_{r2} \end{bmatrix} \tag{37}$$

We can now rewrite (7) expanded in terms of the individual elements of $U_h$, $U_r$, $b_h$, and $b_r$, as shown in (38) and (39).

$$\dot{x} = -\frac{1}{2}[x - \tanh(\frac{U_{h11}x}{1 + e^{-(U_{r11}x + U_{r12}y + b_{r1})}} + \frac{U_{h12}y}{1 + e^{-(U_{r21}x + U_{r22}y + b_{r2})}} + b_{h1})] \tag{38}$$

$$\dot{y} = -\frac{1}{2}[y - \tanh(\frac{U_{h21}x}{1 + e^{-(U_{r11}x + U_{r12}y + b_{r1})}} + \frac{U_{h22}y}{1 + e^{-(U_{r21}x + U_{r22}y + b_{r2})}} + b_{h2})] \tag{39}$$

If 0 is not an eigenvalue of the Jacobian matrix of (38) and (39) at a fixed point, the fixed point is said to be Lefshetz. Let $F_x = \frac{d\dot{x}}{dx}$, $F_y = \frac{d\dot{x}}{dy}$, $G_x = \frac{d\dot{y}}{dx}$, and $F_y = \frac{d\dot{y}}{dy}$.

where

$$F_x = -\frac{1}{2}(1 - \text{sech}^2\left(\frac{U_{h11}x}{e^{-U_{r11}x - U_{r12}y - b_{r1}} + 1} + \frac{U_{h12}y}{e^{-U_{r21}x - U_{r22}y - b_{r2}} + 1} + b_{h1}\right)\gamma_{Fx}) \tag{40}$$

$$\gamma_{Fx} = \frac{U_{h11}U_{r11}xe^{-U_{r11}x - U_{r12}y - b_{r1}}}{(e^{-U_{r11}x - U_{r12}y - b_{r1}} + 1)^2} + \frac{U_{h11}}{e^{-U_{r11}x - U_{r12}y - b_{r1}} + 1} + \frac{U_{h12}U_{r21}ye^{-U_{r21}x - U_{r22}y - b_{r2}}}{(e^{-U_{r21}x - U_{r22}y - b_{r2}} + 1)^2} \tag{41}$$

$$F_y = \frac{1}{2}\text{sech}^2\left(\frac{U_{h11}x}{e^{-U_{r11}x - U_{r12}y - b_{r1}} + 1} + \frac{U_{h12}y}{e^{-U_{r21}x - U_{r22}y - b_{r2}} + 1} + b_{h1}\right)\gamma_{Fy} \tag{42}$$

$$\gamma_{Fy} = \frac{U_{h11}U_{r11}xe^{-U_{r11}x - U_{r12}y - b_{r1}}}{(e^{-U_{r11}x - U_{r12}y - b_{r1}} + 1)^2} + \frac{U_{h12}}{e^{-U_{r21}x - U_{r22}y - b_{r2}} + 1} + \frac{U_{h12}U_{r21}ye^{-U_{r21}x - U_{r22}y - b_{r2}}}{(e^{-U_{r21}x - U_{r22}y - b_{r2}} + 1)^2} \tag{43}$$

$$G_x = \frac{1}{2}\text{sech}^2\left(\frac{U_{h21}x}{e^{-U_{r11}x - U_{r12}y - b_{r1}} + 1} + \frac{U_{h22}y}{e^{-U_{r21}x - U_{r22}y - b_{r2}} + 1} + b_{h2}\right)\gamma_{Gx} \tag{44}$$

$$\gamma_{Gx} = \frac{U_{h21}U_{r11}xe^{-U_{r11}x - U_{r12}y - b_{r1}}}{(e^{-U_{r11}x - U_{r12}y - b_{r1}} + 1)^2} + \frac{U_{h21}}{e^{-U_{r11}x - U_{r12}y - b_{r1}} + 1} + \frac{U_{h22}U_{r21}ye^{-U_{r21}x - U_{r22}y - b_{r2}}}{(e^{-U_{r21}x - U_{r22}y - b_{r2}} + 1)^2} \tag{45}$$

$$G_y = -\frac{1}{2}(1 - \text{sech}^2\left(\frac{U_{h21}x}{e^{-U_{r11}x - U_{r12}y - b_{r1}} + 1} + \frac{U_{h22}y}{e^{-U_{r21}x - U_{r22}y - b_{r2}} + 1} + b_{h2}\right)\gamma_{Gy}) \tag{46}$$

$$\gamma_{Gy} = \frac{U_{h21}U_{r11}xe^{-U_{r11}x - U_{r12}y - b_{r1}}}{(e^{-U_{r11}x - U_{r12}y - b_{r1}} + 1)^2} + \frac{U_{h22}}{e^{-U_{r21}x - U_{r22}y - b_{r2}} + 1} + \frac{U_{h22}U_{r21}ye^{-U_{r21}x - U_{r22}y - b_{r2}}}{(e^{-U_{r21}x - U_{r22}y - b_{r2}} + 1)^2} \tag{47}$$

Let $J$ denote the Jacobian matrix of (38) and (39).

$$J = \begin{bmatrix} F_x & F_y \\ G_x & G_y \end{bmatrix} \tag{48}$$

Note that we can rewrite (38) and (39) as follows:

$$\dot{x} = -\frac{1}{2}[x - \tanh(f(x, y, \theta))] \tag{49}$$

$$\dot{y} = -\frac{1}{2}[y - \tanh(g(x, y, \theta))] \tag{50}$$

where $\theta$ represents the set of parameters, $f(x, y, \theta) = \frac{U_{h11}x}{1 + e^{-(U_{r11}x + U_{r12}y + b_{r1})}} + \frac{U_{h12}y}{1 + e^{-(U_{r21}x + U_{r22}y + b_{r2})}} + b_{h1}$, and $g(x, y, \theta) = \frac{U_{h21}x}{1 + e^{-(U_{r11}x + U_{r12}y + b_{r1})}} + \frac{U_{h22}y}{1 + e^{-(U_{r21}x + U_{r22}y + b_{r2})}} + b_{h2}$

An ordered pair $(x, y)$ is a fixed point of (49) and (50) if and only if (51) and (52) hold.

$$0 = -\frac{1}{2}[x - \tanh(f(x, y, \theta))] \tag{51}$$

$$0 = -\frac{1}{2}[y - \tanh(g(x, y, \theta))] \tag{52}$$

As such, we can say the following:

$$x = \tanh(f(x, y, \theta)) = u(x, y, \theta) \tag{53}$$
$$y = \tanh(g(x, y, \theta)) = v(x, y, \theta) \tag{54}$$

If we let $\lambda$ represent the eigenvalues of (48), the characteristic equation of (48) is as follows:

$$\lambda^2 + \lambda(-1 - \text{sech}^2(f(x, y, \theta))\frac{\partial f}{\partial x} - \text{sech}^2(g(x, y, \theta))\frac{\partial g}{\partial y}) + \frac{1}{4}(1 - \text{sech}^2(f(x, y, \theta))\text{sech}^2(g(x, y, \theta))\frac{\partial f}{\partial y}\frac{\partial g}{\partial x}) \tag{55}$$

We can rewrite (55) in terms of $u(x, y, \theta)$ and $v(x, y, \theta)$ as shown in (56)

$$\lambda^2 + \lambda(-1 - u_x - v_y) + \frac{1}{4}(1 - u_y v_x) \tag{56}$$

where $u_x \equiv \frac{\partial u}{\partial x}$, $u_y \equiv \frac{\partial u}{\partial y}$, $v_x \equiv \frac{\partial v}{\partial x}$, $v_y \equiv \frac{\partial v}{\partial y}$. We can use the quadratic formula to solve for $\lambda$.

$$\lambda = \frac{1 + u_x + v_y}{2} \pm \frac{\sqrt{u_x^2 + 2u_x + 2u_x v_y + 2v_y + v_y^2 + u_y v_x}}{2} \tag{57}$$

Setting $\lambda = 0$ and simplifying yields the following constraint:

$$|u_y v_x| = 1 \tag{58}$$

which can be realized as follows:

$$\text{sech}^2(f(x, y, \theta))\text{sech}^2(g(x, y, \theta)) = \frac{1}{\frac{\partial f}{\partial y}\frac{\partial g}{\partial x}} \tag{59}$$

We observe that $\text{sech}^2(f(x, y, \theta))\text{sech}^2(g(x, y, \theta)) \in (0, 1)$, which implies that $\frac{\partial f}{\partial y}\frac{\partial g}{\partial x} \in (1, \infty)$ However, from (53) and (54), we see that $f(x, y, \theta) = \tanh^{-1}(x)$ and $g(x, y, \theta) = \tanh^{-1}(y)$ at a critical point. Which implies $\frac{\partial f}{\partial y}\frac{\partial g}{\partial x} = 0 \notin (1, \infty)$. Therefore $\lambda \neq 0 \,\forall \theta$.

This implies that (38) and (39) is a Lefschetz map. Since (38) and (39) are asymptotically bound to $(-1, 1)^2$, we can always find a finite time $t_0$ such that $x, y \in (-1, 1)^2 \,\forall t > t_0$. Therefore, for every trajectory initialized outside of the trapping region, we can always find a point on $[-1, 1]^2$ that arises as the transition of that initial condition flowing into the trapping region. This implies that (38) and (39) can be thought of as existing on a compact set, and therefore has a finite number of simple fixed points Guillemin & Pollack (2010).

