# OpenReview forum: "The Expressive Power of Gated Recurrent Units as a Continuous Dynamical System"
_ICLR.cc/2019/Conference_

### Official Review · AnonReviewer2 · 2018-10-25
**Review for the expressive power of GRUs as a continuous dynamical system.**

**Rating:** 5
**Confidence:** 4

**Review:**

Here the authors convert the GRU equations into continuous time and use theory and experiments to study 1- and 2-dimensional GRU networks. The authors showcase every variety of dynamical topology available in these systems and point out that the desirable line and ring attractors are not achievable, except in gross approximation.  The paper is extremely well written.

I am deeply conflicted about this paper.  Is the analysis of 1 or 2 dimensional GRUs interesting or significant? That’s a main question of this paper.  There is no question of quality, or clarity, and I am reasonably certain nobody has analyzed the GRU in this way before.

On the one hand, the authors bring a rigor and language to the discussion of recurrent networks that is both revealing (for these examples) and may to bear fruit in the future.  On the other hand, the paper is exclusively focused on 1- and 2-dimensional examples which have precisely no relevance to the recurrent neural networks as used and studied by machine learning practitioners and researchers, respectively. If the authors have proved something more general for higher dimensional (>2) cases, they should make it as clear as possible.

A second, lesser question of relevance is studying a continuous time version.  It is my understanding that discrete time dynamics may exhibit significantly more complex dynamical phenomenon and again practitioners primarily deploy discrete time GRUs.  I understand that theoretical progress often requires retreating to lower dimensionality and (e.g. linearization, etc.) but in this case it is not clear to me that the end justifies the means.  On the other hand, a publication such as this will not only help to change the language of RNNs in the deep learning community, but also potentially bring in more dynamical systems specialists into the deep learning field, which I thoroughly endorse.

Moderate concern

“In order to show this major limitation of GRUs …” but then a 2-gru is used, which means that it’s not a general problem for GRUs with higher dim, right?  Also, won’t approximate slow points would also be fine here? I think this language needs to be more heavily qualified.

Minor

GRU almost always refers to the network, even though it is Gated Recurrrent Unit, this means that when you write ‘two GRUs’, the naive interpretation (to me) is that you are speaking about two networks and not a GRU network with two units.

Side note requiring no response: It might be interesting to study dynamical portrait as a function of training for the two-d GRU.

---

> ### Author Response · Authors · 2018-11-27
> **re: Review for the expressive power of GRUs as a continuous dynamical system.**
>
> >> Relevance of studying a continuous time version of the GRU
>
> Regardless of dimension, the discrete time GRU-RNN (any RNN or residual network for that matter) can be seen as a forward Euler discretization of an underlying continuous dynamical system, where the topology is dependent on the parameters. As such, there is a relation between the continuous system we derived, and the discrete dynamics of the GRU, which are trying to approximate it. This explanation has also been added to our manuscript.
>
> Moreover, the continuous time limit of residual networks and recurrent neural networks has recently garnered substantial interest. While most of the interest was theoretically motivated, the recent popularization of Neural Ordinary Differential Equations [4], [5]shows the feasibility and usefulness of the continuous-time limit. Our work is highly relevant to these exciting developments and hopefully will provide a useful framework for analyzing the latent representations and, as mentioned in the general introduction, for analyzing the dynamics of the gradients with respect to parameters.
>
> >> “In order to show this major limitation of GRUs …” but then a 2-gru is used, which means that it’s not a general problem for GRUs with higher dim, right? Also, won’t approximate slow points would also be fine here? I think this language needs to be more heavily qualified.
>
> We agree with this concern and have changed the language used in our manuscript to better express what is meant by a limitation, keeping in mind the use of low dimensional latent dynamics, and comparing the results to the more general GRU problem.
>
> The use of a pseudo-line attractor depends on forcing the nullclines to be sufficiently close to one another in order to cause the arbitrarily slow flow in that subregion of phase space to dip below machine epsilon; a more general form of slow point.
>
> However, as a limitation, a general requirement to use approximate slow points in this system (given all functions are smooth, and all parameters can vary smoothly) is that there must exist a separate topology where the slow point in question is a saddle-node bifurcation fixed point (cite bifurcation theory), which limits the number of slow points that can exist for any specific set of parameters.
>
> >> GRU almost always refers to the network, even though it is Gated Recurrent Unit, this means that when you write ‘two GRUs’, the naive interpretation (to me) is that you are speaking about two networks and not a GRU network with two units.
>
> We agree, and have made an explicit note on this point in our updated manuscript as to avoid confusion.

---

### Official Review · AnonReviewer3 · 2018-10-31
**There are several issues with this interesting analysis**

**Rating:** 6
**Confidence:** 4

**Review:**

The authors analyse GRUs with hidden sizes of one and two as continuous-time dynamical systems, claiming that the expressive power of the hidden state representation can provide prior knowledge on how well a GRU will perform on a given dataset. Their analysis shows what kind of hidden state dynamics the GRU can approximate in one and two dimensions. In the experimental part, they show how a GRU with two hidden states trained on a multistep prediction task can learn such dynamics.

Although RNNs are important for Machine Learning, the paper seems to contain flaws in the theoretical part, which seem to invalidate some of the claimed results. But we may change our rating in case of a convincing rebuttal.

The Proof of Lemma 2 claims that h(t) achieves all values on the real set, which is false (h(t) assumes values in (-1,1)). Nevertheless, the theorem should hold since there is always at least one intersection between h and tanh(f(h)).

Lemma 1 claims that for any choice of parameters, there exist only finitely many fixed points. However, in the proof the authors only show that the number of fixed points cannot be uncountable, without taking into consideration the possibility that there are countably many fixed points. The proof also omits steps concerning the Taylor expansion which would make the proof clearer: We suggest adding those steps in the appendix. Furthermore, when equation (12) is Taylor-expanded, the authors do not consider the case where the GRU parameters are such that the argument of function “sech” is outside its convergence radius. These might be parameters for which there are infinitely many fixed points, even if we are unable to provide a Taylor expansion. The Lemma may still be correct, but this does not seem to be a complete proof.

The authors claim that an arbitrarily close approximation of a line attractor can be created using two GRUs, but no proof is provided.

The experimental part is difficult to evaluate since there are no learning curves for the three tasks. For instance, it is difficult to judge whether the GRUs are unable to learn the dynamics of a ring attractor because of theoretical limitations or because the model has not been properly trained for the specific task.

The paper is easy to read, except for certain parts where it is not clear if some of the statements are true in general or just have not been proven false by the authors. It is not clear why Figure 3 is representing all possible simple fixed points and bifurcation fixed points: is there a theoretical result stating that these are the only possible topologies, or are these the only ones found? The same question applies for the 36 images in figure 9. The range of the parameters used for finding these configurations is not specified.

Since the hidden state assumes values in (-1,1)^2, why is its range in most of the images (-1.5,1.5)?

We are not familiar with related work on transformations from discrete to continuous dynamical systems: are the dynamics of the discrete time GRU model preserved in the transformation? If so, is there a reference for this? Are the phase portraits in the middle row of figure 8 generated by letting the discrete GRU system evolve, or is the continuous system used with the parameters of the trained GRU?

We would like to see more explanations of why various topologies are useful for the applications mentioned in the paper. Given a generic dataset, how can these results help to understand how well a GRU will perform?

What is the reason behind the belief that the analysis extends to higher dimensions? The effects of a 1D -> 2D extension are far from trivial - why should that be different for higher dimensions?

The problem the authors want to solve seems important, and some of the theoretical results are promising, but we think that this paper has to be further polished before acceptance.

It is possible that we will increase the score if the authors can provide clarifications on the above questions.

Additional comments:

Introduction

-The vanishing gradient problem was not discovered in 1994, but in 1991 by Hochreiter:

Sepp Hochreiter. Untersuchungen zu dynamischen neuronalen Netzen. Diploma thesis, TU Munich, 1991. Advisor J. Schmidhuber.

- Make clear that GRU is a variant of vanilla LSTM with forget gates (where one gate is missing):

Gers et al. “Learning to Forget: Continual Prediction with LSTM.“ Neural Computation, 12(10):2451-2471, 2000.

- The intro says that GRU has become widely popular and cites Britz et al 2017, but Britz et al actually show that LSTM consistently outperforms its variant GRU in Neural Machine Translation. Please clarify this.

- Also mention Weiss et al (“On the Practical Computational Power of Finite Precision RNNs for Language Recognition”) who exhibited basic limitations of GRU when compared to LSTM.

- Is the result by Weiss et al actually related to the result of the authors who found that 2 GRUS cannot accurately a line attractor without near zero constant curvature in the phase space?


Section 2

-Wrong brackets in equation (4)
-Missing bracket before citing Laurent & Brecht

Section 4

-”We conjecture that the system depicted in figure 2A..” Should be figure 3A
- Lemma1: UZ has capital Z subscript

Section 5.2

-”The added affine transformation allows for a sufficiently long subinterval”: “sufficiently long” is too vague

Section 5.3

“A manifold with without near zero constant curvature”: should be “a manifold without near zero constant curvature”

Appendix A

-Wrong brackets in equation (20)

Appendix B

- In the proof of Theorem 1, the derivative is of (29), not of (12)

Appendix C

Figure 9: “who’s initial conditions” should be “whose initial conditions”

After rebuttal:

It's better now. However, the revised introduction still says:  "GRU has become wildly popular in the machine learning community thanks to its performance in machine translation (Britz et al., 2017) ... LSTM has been shown to outperform GRU on neural machine translation (Britz et al., 2017).... specifically unbounded counting, come easy to LSTM networks but not to GRU networks (Weiss et al., 2018)."

So better remove the first statement on Britz et al: "GRU has become wildly popular ... in machine translation (Britz et al., 2017)" because they actually show why GRU is NOT wildly popular in machine translation, as correctly justified later in the same paragraph.

Pending the above revision, we'd like to increase our evaluation by 2 points, up to 6!

---

> ### Author Response · Authors · 2018-11-27
> **re:There are several issues with this interesting analysis**
>
> Reviewer Specific Concerns:
>
> >> The Proof of Lemma 2 claims that h(t) achieves all values on the real set, which is false (h(t) assumes values in (-1,1)).
>
> We’ve reworked this proof to avoid previous confusion. h(t), while asymptotically bounded to (-1,1), can be initialized anywhere on the reals, whereby it will eventually fall into the designated trapping region. Since h(t) can be realized as line of unit slope, it is unbounded and bijective, obtaining all values on the reals.
>
> >> Lemma 1 does not seem like a complete proof.
>
> Lemma 1 has been rewritten in its entirety as a means to (1) improve readability, and (2) include the countably infinite case. The authors believe this approach is better suited for the paper, as proof for the countable case is contained within the argument, and no information is left out, as Taylor series approximation is not used. Moreover, we have now extended the proof to arbitrary dimensions using differential geometry arguments. These are the new Theorem 1 and 2.
>
> >> The authors claim that an arbitrarily close approximation of a line attractor can be created using two GRUs, but no proof is provided.
>
> We apologize for the confusion on this concern. We show by existence that a 2D GRU can approximate a straight (or nearly straight) line attractor. However, an arbitrary line attractor cannot be mimicked to machine precision. We’ve adjusted the language of the paper to avoid misinterpretation.
>
> >> The experimental part is difficult to evaluate since there are no learning curves for the three tasks. For instance, it is difficult to judge whether the GRUs are unable to learn the dynamics of a ring attractor because of theoretical limitations or because the model has not been properly trained for the specific task.
>
> We have extended our experiments to illustrate the inability of the 2D GRU to capture the dynamics of a ring attractor. We compared the k-step MSE as a function of number of epochs and as a function of latent dimensionality. We observe that for the ring attractor the MSE decreases as the latent dimensionality increases. On the other hand, the MSE for the FitzHugh-Nagumo does not decrease appreciably as the latent dimensionality increases.
>
> >> The paper is easy to read, except for certain parts where it is not clear if some of the statements are true in general or just have not been proven false by the authors. It is not clear why Figure 3 is representing all possible simple fixed points and bifurcation fixed points: is there a theoretical result stating that these are the only possible topologies, or are these the only ones found? The same question applies for the 36 images in figure 9. The range of the parameters used for finding these configurations is not specified.
>
> Thank you for bringing our attention to this. Indeed our language was ambiguous at places. We have updated the language we use in order to make these points clearer. Generally speaking, the 1D analysis is exhaustive, as nothing beyond what we show exists. Figure 3 depicts all local structures found by the authors. Similarly, figure 9 depicts all global structures found by the authors.
> The range of parameters was not specified as no set range was considered in discovering these structures. Rather, all structures were found by a combinatorial systematic procedure, by means of considering all possible ways the nullclines can intersect, given their geometric structure. We’ve added an explanation of this method in the appendix.
>
> >> Since the hidden state assumes values in (-1,1)^2, why is its range in most of the images (-1.5,1.5)?
>
> We agree that all interesting/practical behavior takes place in this bounded region in 2D. However, in section 2 it is stated that “the hidden state is asymptotically contained within [-1,1]^d.” As a result, the hidden state only assumes values in (-1,1)^2 if and only if initialized within that region of phase space. The range of (-1.5,1.5) was used to better improve visualization of global dynamics, by including trajectories initialized outside the trapping region. This point is further emphasized in the updated paper.

---

> ### Author Response · Authors · 2018-11-27
> **cnt'd**
>
> >> We are not familiar with related work on transformations from discrete to continuous dynamical systems: are the dynamics of the discrete time GRU model preserved in the transformation? If so, is there a reference for this?
>
> A GRU-RNN (more generally any RNN or residual network) can be seen as a forward Euler discretization of an underlying continuous time dynamical system (see [4] and references within). As such, the discrete and continuous time systems have very similar forms, as the GRU is attempting to approximate the system we analyzed. The continuous-time dynamical systems framework preserves the smooth temporal structures and ignore the possible quirky/jumpy features of discrete maps which powers our analysis. However, generally speaking, the dynamic
> properties are not always preserved when converting from discrete to continuous time. For example [3] showed that the 2D discrete GRU can exhibit chaos. However, a 2D continuous time dynamical system cannot show signs of chaos, a result of the Poincare-Bendixson theorem (J. Meiss. Differential Dynamical Systems).
>
> >> Are the phase portraits in the middle row of figure 8 generated by letting the discrete GRU system evolve, or is the continuous system used with the parameters of the trained GRU?
>
> The dynamics shown in the middle row of figure 8 are those of the trained GRU on the continuous time system.
>
> The reviewer then lists out a series of additional comments. We have gone through each individually and made the suggested corrections.
>
> >> Is the result by Weiss et al actually related to the result of the authors who found that 2 GRUS cannot accurately approximate a line attractor without near zero constant curvature in the phase space?
>
> Yes, the mechanistic act of counting in the [Weiss et al.] paper using LSTMs has a continuous time analog to a line attractor, with a forcing term propelling the state parallel to the attractor. Since the GRU lacks an output gate, its hidden state acts as a hybrid between the LSTM’s cell state and output state. As a result, the GRU hidden state must exist asymptotically on a compact set, which is not true for the LSTM cell state. This limitation is necessary in proving that the finite dimensional GRU cannot exhibit a line attractor.

---

### Official Review · AnonReviewer1 · 2018-11-06
**a dynamical systems analysis of 1d and 2d gated recurrent units**

**Rating:** 5
**Confidence:** 4

**Review:**

This paper analyzes GRUs from a dynamical systems perspective, i.e. phase diagrams, fixed points, and bifurcations. The abstract and intro are well written and motivate the need for more theoretical framework to understand RNNs, especially how well they are able to represent and express temporal features in the training data. The dynamical systems analysis is well presented and visualized nicely.

Most of the paper concentrates on the 1d (one single GRU) and 2d (two GRU's) case.  They show that 2d GRUs can be trained to adopt a variety of fixed points, can approximate a line attractors (an important feature for short-term memory), but cannot mimic a ring attractor.

My concerns are:

- The derivation of the continuous time dynamical system (Appendix A) is confusing to me. Unless I'm not following the derivation correctly, should there be another \Delta t in the denominator of the right-hand side of (23), from (22)? It's confusing to me that the continuous-time version in (26) has essentially the same form as the discrete-time version in (22).

- The applicability of this analysis to RNNs of even modest size is unclear. Generically, there's no reason to believe the intuitions from 2d should necessarily generalize to higher dimensions, and rigorous analysis of higher dimensional systems of this kind can be fairly challenging, even if one starts from a continuation analysis.

- Small typo: top of Page 4, figure should refer to 3A, not 2A.

---

> ### Author Response · Authors · 2018-11-27
> **re: a dynamical systems analysis of 1d and 2d gated recurrent units**
>
> Reviewer Specific Concerns:
>
> >> Readability and validity of the continuous time system derivation.
>
> We’ve added several intermediate steps to the derivation (Appendix A) as a means of improving readability. Note that a GRU-RNN (any RNN or residual network for that matter) can be seen as a forward Euler discretization of an underlying continuous time dynamical system. Under this discretization, the derivative with respect to time appears on the right-hand side of the equation 23, implying both the continuous and discrete time systems are of the same functional form. Furthermore, to clarify the reason for the seemingly missing $\delta t$, let us point out that the Euler discretization is valid for general ODEs with arbitrary step sizes. In the GRU it’s implicitly assumed that $\delta t = 1$, which is why the time step doesn’t appear in the GRU update equation for $h_{t+1}$. In the derivation we made use of this fact, but failed to make it explicit. We amend it in the new version of the manuscript.
>
> >> Small typo on the top of page 4
> We have corrected it.

---

### Author Response · Authors · 2018-11-27
**General response to all reviewers**

We thank the reviewers for their careful reading of our manuscript and many helpful suggestions. We are flattered that the reviewers found the manuscript well written and original.

First, we would like to briefly emphasize the importance of our work. Originally, GRUs were designed to mitigate the difficulty of training recurrent neural networks on tasks with long temporal dependence. In their ingenious use of different gates both LSTMs and GRUs were believed to store information until it is needed at a later time. However, prior to our analysis, little was formally known about how hidden states store information in their dynamics. We extend this understanding by exhaustively listing the types of dynamics that GRU network can generate. These include stable limit cycles over time (nonlinear oscillations), multi-stable state transitions with various topologies, or generating stereotypical temporal responses to perturbations (homoclinic orbit). We were pleasantly surprised to discover this rich expressive power of the 2D GRU system. This was possible thanks to the continuous-time dynamical systems framework that allows us to focus on the smooth temporal structures and ignore the possible quirky/jumpy features of discrete maps.

Furthermore, the analysis of 1-D and planar hidden state dynamics offers a new approach to the analysis of recurrent neural networks. Existing approaches have chose other simplifications, ranging from the analysis of linear dynamics to mean field approaches [1], [2]. The latter has extended our understanding of the dynamics of large, randomly initialized networks. The approach championed in this work, considers, in a sense, the opposite simplification to that used in the mean field analysis. Here we have considered networks with 1 or 2 neurons in the hidden layer, but have derived the classes of dynamics that these simple networks always fall into, both at random initialization and throughout training. We believe that our analysis will be helpful in the future in deepening our understanding of the learning dynamics since the backward pass (for backpropagating gradients) requires computing a linearization of the forward dynamics. The dynamical systems perspective is intimately connected to learning, since the stability of equilibria is measured with the eigenvalues of the same linearization. Therefore understanding the topological properties of the forward dynamics gives insight into the topology of learning  dynamics. In future work we hope to leverage this connection to better understand gradient dynamics during learning.

Major changes:
We replaced previous claims about point and line attractors to arbitrary dimensions. (Theorem 1 & 2)
New experiments showing learning curves for higher dimensional GRU networks (Fig. 9)
Overall improvement in writing.

References:
[1]M. Hardt, T. Ma, and B. Recht, “Gradient Descent Learns Linear Dynamical Systems,” J. Mach. Learn. Res., vol. 19, no. 29.
[2]M. Chen, J. Pennington, and S. Schoenholz, “Dynamical Isometry and a Mean Field Theory of RNNs: Gating Enables Signal Propagation in Recurrent Neural Networks,” in International Conference on Machine Learning, 2018, pp. 873–882.
[3]T. Laurent and J. von Brecht, “A recurrent neural network without chaos,” Nov. 2016.
[4]R. T. Q. Chen, Y. Rubanova, J. Bettencourt, and D. Duvenaud, “Neural Ordinary Differential Equations,” ArXiv180607366 Cs Stat, Jun. 2018.
[5]W. Grathwohl, R. T. Q. Chen, J. Bettencourt, I. Sutskever, and D. Duvenaud, “FFJORD: Free-form Continuous Dynamics for Scalable Reversible Generative Models,” ArXiv181001367 Cs Stat, Oct. 2018.

---

### Meta-Review · Area_Chair1 · 2018-12-12
**scaling issue**

**Confidence:** 4
**Recommendation:** Reject

**Metareview:**

The paper analyses GRUs using dynamic systems theory.  The paper is well-written and the theory seems to be solid.

But there is agreement amongst the reviewers that the application of the method might not scale well beyond rather simple 1- or 2-D GRUs (i.e., with one or two GRUs).  This limitation, which is an increasingly serious problem in machine-learning papers, should be solved before the paper should be published.  A very recent extension of the simulations to 16 GRUs improves this, but a rigorous analysis of higher-dimensional systems is pending and poses a considerable block for acceptance.